



# 1 Snow cover variations across China from 1951-2018

Xiaodong Huang[1*], Changyu Liu[2], Zhaojun Zheng[3], Yunlong Wang[2], Xubing Li[1], and Tiangang Liang[2]

[1] School of Geographical Sciences, Nanjing University of Information Science and Technology, Nanjing 210044, China

[2] State Key Laboratory of Grassland Agro-ecosystems, College of Pastoral Agriculture Science and Technology, Lanzhou University, Lanzhou, 730020, China

[3] Key Laboratory of Radiometric Calibration and Validation for Environmental Satellites, National Satellite Meteorological Center, China Meteorological Administration, Beijing, 100081, China

*Correspondence to*: Xiaodong Huang (huangxd@lzu.edu.cn)

**Abstract.** Based on a snow depth dataset retrieved from meteorological stations, this experiment explored snow indices, including snow depth (SD), snow covered days (SCDs), and snow phenology variations, across China from 1951 to 2018. The results indicated that the snow cover trends across China exhibits regional differences. The annual mean SD tended to increase, and the increases in mean and maximum snow depth were 0.04 cm and 0.1 cm per decade, respectively. SCDs tended to increase by approximately 0.5 days per decade. The significant increases were concentrated at latitudes higher than 40°N, especially in Northeast China. However, in the Tibetan Plateau, the SD and SCDs tended to decrease but not significantly. Regarding the snow phenology variations, the snow duration days in China decreased, and 25.2% of the meteorological stations showed significant decreasing trends. This result was mainly caused by the postponement of the snow onset date and the advancement of the snow end date. Geographical and meteorological factors are closely related to snow cover, especially the change in temperature, which will lead to significant changes in snow depth and phenology.

Keywords: Snow; Ground observations; Change; China

## 1 Introduction

Snow covers 40% of the global land surface in winter, and more than 90% of seasonal snow cover is concentrated in the Northern Hemisphere (Armstrong and Brodzik, 2001), stretching across an area of approximately $4.6 \times 10^7$ km$^2$. Snow cover represents an essential component of the energy exchange process and hydrological cycle within the global climate system (Euskirchen et al. 2007; Yao et al. 2013). Snow cover has a unique physical attribute of high albedo (Xiao and Che, 2016), which has a positive feedback effect on climate (Tedesco and Miller, 2007). Within the global hydrological cycle, snow cover not only affects the water cycle but also constitutes a highly crucial form of water storage (Ambadan, 2017; Shams et al., 2018). However, snow can also have negative impacts on human life because snowfall and meltwater are direct causes of snowmelt erosion, snowmelt floods, avalanches, and other natural disasters (Li & Simonovic, 2010; Chen et al., 2016).

As an essential climate variable, snow cover has received much attention around the world, and various snow datasets can be used to evaluate snow cover variations. Currently, the Rutgers University Global Snow Lab and the binary snow cover mask data derived from the Climate Data Record of the Northern Hemisphere Snow Cover Extent (NHSCE) can provide a long-term snow dataset (1967-present). However, the dataset is appropriate for only large-scale snow extent studies because of the coarse spatial resolution (24 km), much coarser for the 1967-1998 portion of the record (190.5 km resolution) (Brown & Robinson, 2011). In addition, the implementation of the interactive multi-sensor snow and ice mapping system (IMS) provides another approach for the dynamic monitoring of snow extent (Sönmez et al. 2014).



Other sensors with moderate resolution, such as Moderate Resolution Imaging Spectroradiometer
(MODIS), can provide global snow extent products with high resolution and accuracy, but the record
period is short (2000 to present) (Hall et al., 2002). Passive microwave remote sensing has been regarded
as an efficient way to retrieve snow depth (SD) or snow water equivalent (SWE) data at hemisphere and
global scales, such as the Scanning Multichannel Microwave Radiometer (SMMR), Special Sensor
Microwave/Imager (SSM/I), and Advanced Microwave Scanning Radiometer-EOS (AMSR-E). Another
technique that assimilates in situ snow depth observations with microwave emissions was applied in the
European Space Agency's (ESA) GlobSnow project to estimate the daily SWE time series from 1979 to
present over the Northern Hemisphere (GlobSnow v3.0 SWE CDR released in 2019), and this technique
is considered to overcome the large errors that rely solely on passive microwave observations (Pulliainen
et al. 2006; 2020; Takala et al., 2011). However, while the GlobSnow SWE algorithm exhibits improved
accuracy, the data gaps in alpine areas limit its comprehensive use in snow variation assessments.
13       Early satellite observations in the Northern Hemisphere suggested a decline in the snow cover extent
(SCE) over the past several decades, even though the spatial trend patterns are variable between different
observations. The SCE has experienced a well-documented decrease in spring, and arguably, there has
been a slight increase or decrease in winter (the statistical significance of the linear trend is very weak)
in satellite observations since the late 1980s in the Northern Hemisphere (Brown and Robinson, 2011;
Choi et al, 2010; Cohen et al. 2014; Connolly et al. 2019; Mudryk et al. 2020). The pattern of the remote
sensing observations is quite different from the Coupled Model Intercomparison Project Phase 5 (CMIP-
5) predictions, in which the CMIP5 models imply that SCE have steadily decreased for all seasons.
(Connolly et al. 2019). The spring SCE that was determined using the NHSCE between 1967 and 2012
over the Northern Hemisphere was compared with the CMIP-5 model output, which revealed that the
reductions in the spring SCE from 2008–2012 exceeded the CMIP-5 projections (Derksen and Brown,
2012). Multi-dataset analysis shows decrease in SCE in all months from 1981 to 2014 in Northern
Hemisphere, and the CMIP-6 multi-model ensemble better represents the snow extent climatology for
all months, correcting a low bias in CMIP-5 (Mudryk et al. 2020). During the period from 2000 to 2015,
the SCE in high-latitude and high-elevation mountainous regions decreased significantly, while the SCE
at some middle and low latitudes showed increasing trends in the Northern Hemisphere (Wang et al.,
2018). In Eurasia, the SCE decreased significantly only in June, and there is no obvious trend of SCE in
winter from 2000 to 2016 (Sun et al. 2020). A delayed snow onset date was observed to be the main
driver of decreasing annual snow duration trends, and the spatial pattern of annual snow duration trends
exhibited noticeable asymmetry between continents, with the largest significant decreases observed over
western Eurasia with relatively few statistically significant decreases over North America (Hori et al.,
2017). This finding shows the regional differences from the hemispheric average trends for the Northern
Hemisphere (Brown & Robinson, 2011; Wang et al., 2018; Connolly et al., 2019). In contrast to SCE,
the annual mean SD decreased in most areas over North America (Dyer & Mote, 2006) and increased in
Eurasia and the Arctic (Kitaev et al., 2005; Liston & Hiemstra, 2011). Accordingly, the overall snow
mass trends were negative over North America, and for Eurasia, the trend was negligible for the
investigated non-alpine regions from 1980 to 2018 (Pulliainen et al., 2020).
40       Limited by the coarse spatial resolution, poor accuracy and short observation periods from remote
sensing data, in situ snow depth observations provide the most reliable dataset for analyzing the changes
in snow cover with a high degree of credibility. Moreover, snow parameters are calculated from
meteorological station data, which have great advantages in the process of long time series research.
Measurements of daily snow depth were conducted at 1103 meteorological stations over the Eurasian



continent from 1966 to 2012 to provide a detailed description of snow depth variability. These measurements revealed that both the annual mean and the maximum SD showed increasing trends over the entire Eurasian continent, including China, and the snow depth decreased in autumn and increased in spring and winter (Zhong et al. 2018). Daily snow observation data from 672 stations in China during the period from 1952–2010 were used to analyze snow cover days (SCDs) and snow cover phenology variations (Ke et al. 2016). The results indicated that from 1952 to 2010, the overall snow phenology in China reflected a delay of the snow onset date (SOD) and an advancement of the snow end date (SED). The reduced temperature and increased mean air temperature were the main reasons for the overall late snow onset and early snow end day.

In recent years, the variations in snow cover over China have attracted much attention, especially with regard to the so-called 'third pole' of the Tibetan Plateau due to it being the region with the highest elevation and deepest snow depth at middle latitudes in the Northern Hemisphere (Ma et al., 2010). A previous study using remote sensing snow products indicated that the annual mean SCE accounts for 27% of the country's total area in winter, and the average annual SCE decreased during winter and summer but increased in spring and fall over the last 14 years; however, these trends were not statistically significant (Huang et al., 2016). Driven by decreased temperature and increased precipitation in the snow accumulation season, the snow cover fraction over mainland China showed an increasing trend of 0.29% per decade during 1982–2013, which was significant at the 0.05 level (Chen et al. 2016). Overall, the SCE varied slightly but did not increase or decrease significantly for the Chinese mainland. The SOD moved forward slightly, but the SED has become significantly earlier at the rate of 1.91 days per decade, with a 73% contribution from the decreased SCE between 1982 and 2013 in China (Chen et al., 2016).

SD is a basic and important parameter of snow cover that plays an important role in hydrological applications, numerical weather predictions, climate change research and land surface process simulations. However, reliable quantitative knowledge of long-term seasonal snow cover and its trend is lacking. Therefore, we aim to explore the snow cover variations in China from 1951 to 2018 based on an SD dataset retrieved from meteorological stations. The objectives are to 1) evaluate the spatial distributions of various snow parameters, 2) ascertain the variation trends and fluctuation periods of those snow parameters, 3) compare the trends of those snow parameters among the three stable snow cover areas of China, and 4) analyze the spatiotemporal dynamics of snow heterogeneity by using geographical and meteorological factors throughout China.

## 2 Dataset and methodology

### 2.1 Snow depth records

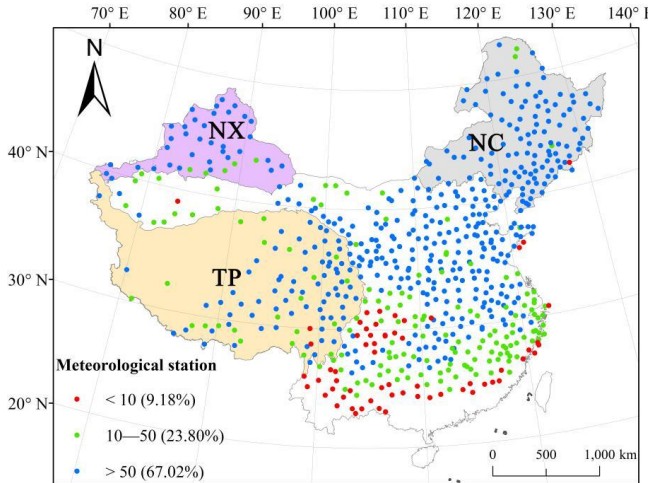

**Figure 1.** Geographical locations and the proportion of valid yearly records between 1951 and 2018 for
each meteorological station in mainland China. The abbreviations of snow cover areas represent the
Tibetan Plateau (TP), northern Xinjiang (NX), and Northeast China (NC).
Snow depth datasets were obtained from 730 meteorological stations from January 1, 1951, to December
31, 2018, in China, which were provided by the National Meteorological Information Center of the China
Meteorological Administration (CMA) (http://data.cma.cn/en). A hydrological year spanned from July 1
of the current year to June 30 of the subsequent year to capture the entire seasonal snow cycle. Therefore,
the data from July 1, 1951, to June 30, 2018, were used in this study, which include 67 seasonal snow
cycles in total (Figure 1). Snow depth of ground observations is measured manually with a wooden ruler
at 8 o'clock every day when the ground in the field of view around the meteorological station is covered
by more than half in snow. The measurements are made thrice and the distance between the three
measurements is more than 10 m. The measured value is accurate to 0.1 cm. The final snow depth at each
station was determined as the average of the three measurements, and an average snow depth of less than
0.5 cm is recorded as 0. Initially, the data quality control standards implemented in this study were as
follows. 1) Only daily SD values larger than 1 cm were recorded as snow cover; stations with SD values
less than 1 cm were regarded as snow free. 2) Stations with records spanning less than 10 years were
omitted from the analysis to ensure the reasonableness of the statistical analysis. Therefore, 67 stations
with snow records less than 10 years were omitted from the analysis in this study.
The snow cover parameters, including the annual mean snow depth (SD$_{overall}$), maximum snow depth
(SD$_{max}$), SCDs, SOD, SED, and snow duration days (SDDs), were calculated for each selected
meteorological station (Table 1). To avoid the impact of ephemeral snow in snow phenology
computations, SOD was defined as the first date of the first three continuous snow records, and SED was
defined as the last day of the date of last three continuous snow records (Chen et al. 2016; Ke et al. 2016).
The difference between the numbers of SCDs and SDDs is that SCDs include ephemeral snow days
beyond the snow season, which may be longer than SDDs.
**Table 1**. The abbreviations and descriptions of snow parameters

| Abbreviation | Description |
| --- | --- |
| SD$_{overall}$ | Calculated by dividing the sum of snow depth records by the total number of days in a hydrological year |





| SD$_{max}$ | The maximum snow depth for the corresponding hydrologic year |
|---|---|
| SCDs | The total days characterized by snow-covered ground throughout a hydrological year |
| SOD | The first date of snow onset in the snow accumulation season during a hydrological year |
| SED | The snow end date during the snow melting season |
| SDDs | The number of days from the SOD to the SED in a corresponding hydrological year |

**2.2 Meteorological data**

Meteorological data with a 1 km resolution were selected in this study. The dataset was provided by the "Loess Plateau Data Center, National Earth System Science Data Sharing Infrastructure, National Science & Technology Infrastructure of China (http://loess.geodata.cn)". The dataset time series is from 1901 to 2017. In this study, the annual mean temperature (Tmp-mean), annual total precipitation (Pre-sum), annual maximum monthly precipitation (Pre-max), annual mean lowest temperature (Tmn-mean), annual mean maximum temperature (Tmx-mean), annual coldest monthly minimum temperature (Tmn-min), and annual warmest monthly maximum temperature (Tmx-max) from 1951 to 2017 in the dataset were used to explore the heterogeneity of snow cover.

**3 Methodology**

**3.1 Trend analysis**

Linear fitting is the most common and extensive trend analysis method. Moreover, the Mann-Kendall (M-K) test is also recommended by the World Meteorological Organization and is frequently used to analyze the trends of changes in meteorological and hydrological elements (Milan, 2013). In this study, these two methods were used to analyze the trends of the variations in the snow cover indices from 1952 to 2012. The M-K test formulas are as follows:

$$S = \sum_{j=1}^{n-1} \sum_{i=j+1}^{n} sign(x_i - x_j) \tag{1}$$

where $n$ is the number of years to be analyzed. $x_i$ and $x_j$ are the values in time series $i$ and $j$, respectively.

$$\begin{cases} Z = \dfrac{(S-1)}{\sqrt{\dfrac{n(n-1)(2n+5)}{18}}} & S > 0 \\ Z = 0 & S = 0 \\ Z = \dfrac{(S+1)}{\sqrt{\dfrac{n(n-1)(2n+5)}{18}}} & S < 0 \end{cases} \tag{2}$$

where $Z$ is the value used to judge whether the trend is increasing or decreasing in the trend analysis. When $Z$ is positive, the trend is increasing, while negative values of $Z$ represent decreasing trends. At the same time, by comparing the absolute value of $Z$ with the standard value of $Z$, we can determine the significance of the trend. In this study, significance levels of α=0.05 and α=0.01 are used. If the absolute value of $Z$ is greater than $Z_{0.05}$ or $Z_{0.01}$, the trend is statistically significant or extremely significant, respectively.

$$S_k = \sum_{i=1}^{k} r_i, r_i = \begin{cases} 1, x_i > x_j \\ 0, x_i \le x_j \end{cases}, (j = 1,2,\dots,i; k = 1,2,\dots,n) \tag{3}$$

$$E[S_k] = \frac{k(k-1)}{4} \tag{4}$$

$$var[S_k] = \frac{k(k-1)(2k-5)}{72} \quad 1 \le k \le n \tag{5}$$

$$UF_k = \frac{(S_k - E[S_k])}{\sqrt{var[S_k]}} \tag{6}$$





$UB_k = -UF'_k, UF'_k = UF_{n-k}$ (7)
where $UF$ is the standardized value of $S$, while $UF'$ is obtained by inverting the sequence of $UF$.
In the M-K test, when the $UF$ is greater than 0, there is an increasing trend from the initial year to
the corresponding year; when the $UF$ is less than 0, there is a decreasing trend. Similarly, when $UB$ is
greater than 0, there is an increasing trend from the corresponding year to the end year, and when $UB$ is
less than 0, there is a decreasing trend. $UF$ and $UB$ can provide the approximate break points of the
meteorological sequence. However, the break points of the meteorological sequence can be further
judged by combining the M-K and moving t tests. When $t$ is greater than $t_{0.05}$, the year corresponding to
$t$ represents a break point. The formulas for the moving $t$ test are as follows:
$t = \dfrac{(\bar{x}_1 - \bar{x}_2)}{s\sqrt{\frac{1}{n_1} + \frac{1}{n_2}}}$ (8)
$s = \sqrt{\dfrac{n_1 s_1^2 + n_2 s_2^2}{n_1 + n_2 - 2}}$ (9)
In this study, the slope method is also employed to analyze the snow cover variation trend (Stow et al.
2004). The formula is as follows:
$\text{slope} = \dfrac{n\sum_{i=1}^{n} i x_i - \sum_{i=1}^{n} i \sum_{i=1}^{n} x_i}{n\sum_{i=1}^{n} i^2 - (\sum_{i=1}^{n} i)^2}$ (10)
where $n$ is the number of datasets to be analyzed and $x_i$ is the value in the time series $i$.
**3.2 Structural equation model**
Structural equation modeling (SEM) is a statistical method based on a variable covariance matrix
that can be used to analyze the relationships between variables (Bagozzi and Yi, 2012). This method
synthesizes a variety of statistical methods, including path analysis, regression analysis, and factor
analysis. Path analysis is a method to analyze the multilayer relationship and correlation intensity
between multiple variables. In this study, path analysis was employed to analyze the effect of
geographical and meteorological factors on the spatial and temporal heterogeneity of snow cover. To
avoid collinearity within the variables, a factor importance screening function was carried out for variable
selection. Finally, seven factors including altitude, latitude, longitude, Tmp-mean, Pre-sum, Tmx-max,
and Tmn-min were screened out. Conceptual models of the hypothesized relationships showing the direct
and indirect effects of geographical and meteorological variables on the snow parameters are shown in
Figure 2. The standardized path coefficients for SME assume that latitude, longitude and altitude directly
affect precipitation and temperature and thus indirectly affect snow cover, while precipitation and
temperature have a direct impact on snow cover. The maximum likelihood algorithm was used to estimate
the model parameters and determine the goodness-of-fit of the model (Grace et al., 2010). The model
output of the path coefficients are standardized regression coefficients. The total standardized effects
consist of direct and indirect standardized effects. High path coefficients indicate large effects of a
predictor variable on the response variable (Grace et al., 2016). The formula for the maximum likelihood
method is as follows:
$F_{ML} = \log|\Sigma(\theta)| + tr\left(S\sum(\theta)^{-1}\right) - \log|S| - (p + q)$ (11)
$S = \begin{pmatrix} var(y) & cor(y,x) \\ cor(x,y) & var(x) \end{pmatrix}$ (12)
$\Sigma(\theta) = \begin{pmatrix} var(x) + var(e) & var(x) \\ var(x) & var(x) \end{pmatrix}$ (13)
where the core of the maximum likelihood method is to minimize the function $F(S, \sum(\theta))$. $S$ is the
sample covariance matrix, and $\sum(\theta)$ is the implicit covariance of the structural variable.

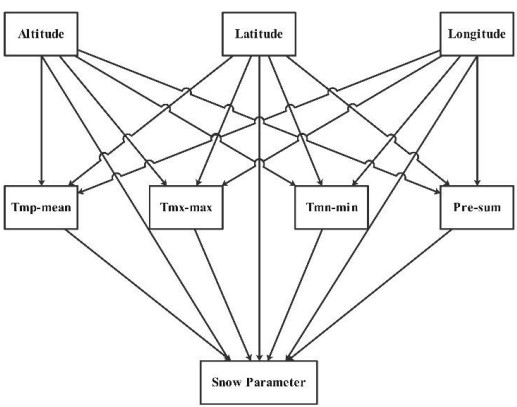

**Figure 2.** Conceptual models of the hypothesized relationships showing the direct and indirect effects
of geographical and meteorological variables on snow parameters
**4    Results**
**4.1   SD**
The mean annual $SD_{overall}$ and $SD_{max}$ gradually increase with increasing latitude and altitude in China
from 1951 to 2018 (Figure 3a and c). The largest mean $SD_{overall}$ of 9.3 cm is found in northern Xinjiang,
while the largest annual mean $SD_{max}$ (55.3 cm) appears in the Tibetan Plateau. Overall, the mean $SD_{overall}$
and $SD_{max}$ are 0.7 cm and 9.4 cm, respectively. $SD_{overall}$ and $SD_{max}$ both tend to increase by 0.05 cm and
0.1 cm per decade from 1951 to 2018, respectively. Additionally, the distributions of the $SD_{overall}$ and
$SD_{max}$ trends are similar (Figure 3b, and d). The stations in China with significant increases in $SD_{overall}$
and $SD_{max}$ are concentrated mainly at high latitudes, and the proportions of meteorological stations with
significant increases in $SD_{overall}$ and $SD_{max}$ are 9.8% and 7.3%, respectively. In contrast, the stations with
significant decreases in $SD_{overall}$ and $SD_{max}$ are concentrated mainly in the Tibetan Plateau and central
China, and the proportions of meteorological stations with significant decreases in $SD_{overall}$ and $SD_{max}$ are
7.1% and 5.6%, respectively. Table 2 also shows that the overall $SD_{overall}$ and $SD_{max}$ tend to increase
significantly in Northeast China, increasing by 0.2 cm and 0.7 cm per decade, respectively. These values
in northern Xinjiang tend to increase but not significantly. In the Tibetan Plateau, however, $SD_{overall}$ and
$SD_{max}$ both show decreasing trends, but the trends are negligible.
**Table 2** Trends in SD across the three snow cover areas of China from 1951 to 2018.

| Zone | Variate | Slope analysis | | M-K analysis |
|---|---|---|---|---|
| | | Slope | *P*-value | Z-value |
| Northeast | $SD_{overall}$ | 0.02 | 0.00** | 3.51** |
| | $SD_{max}$ | 0.07 | 0.00** | 3.00** |
| Northern Xinjiang | $SD_{overall}$ | 0.01 | 0.15 | 1.22 |
| | $SD_{max}$ | 0.04 | 0.14 | 1.68 |
| Tibetan Plateau | $SD_{overall}$ | -0.00 | 0.78 | -0.13 |
| | $SD_{max}$ | -0.00 | 0.79 | -0.02 |

** denotes a significant change at $P < 0.05$.

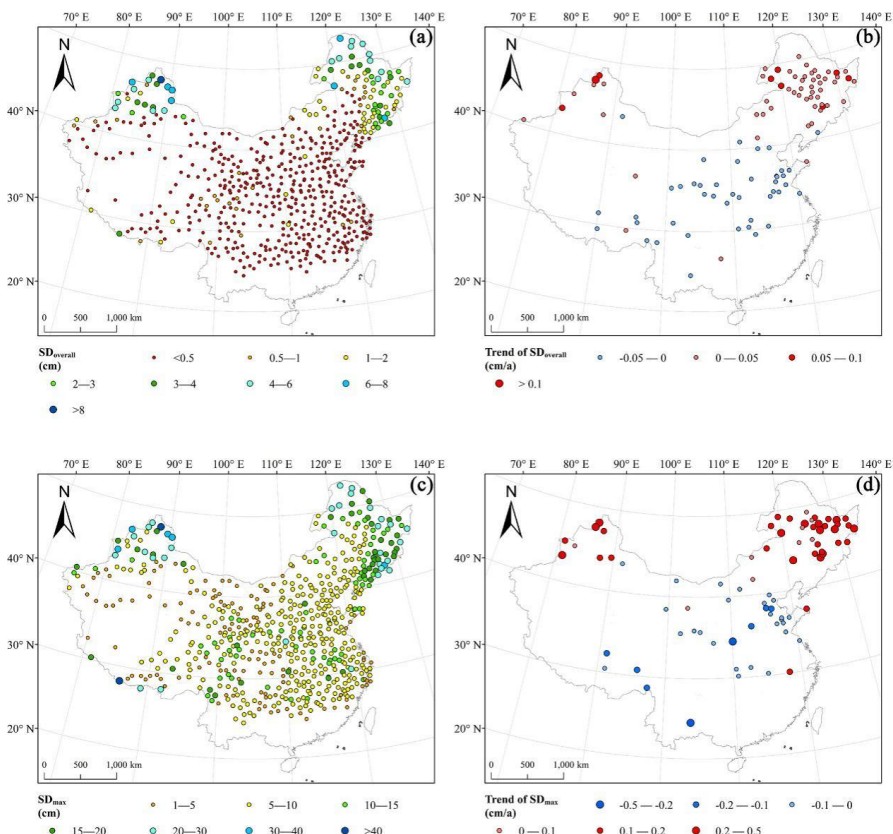

**Figure 3.** Panels (a) and (c) represent the spatial distributions of the mean annual $SD_{overall}$ and the mean
$SD_{max}$ across China, respectively, and panels (b) and (d) show the distributions of the trends of the mean
annual $SD_{overall}$ and the mean $SD_{max}$, respectively, as determined by the trend analysis to exhibit
significant changes ($P < 0.05$) across China.

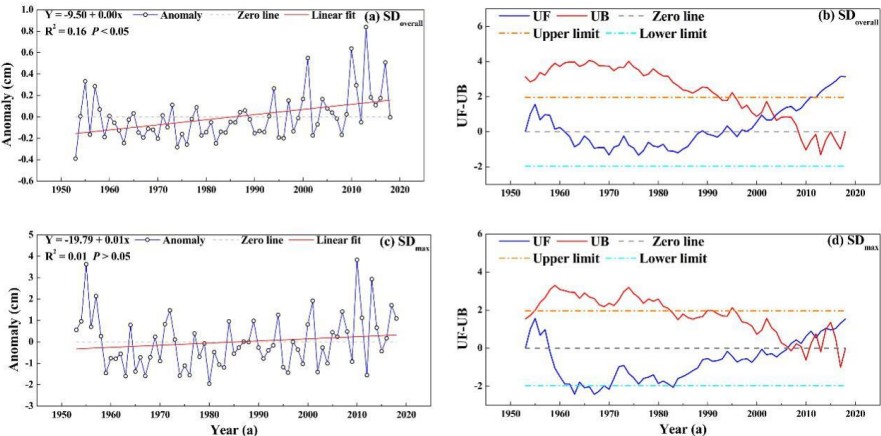





**Figure 4.** Panels (a) and (c) present the linear fits of the mean annual $SD_{overall}$ and the mean $SD_{max}$ across
China, respectively, and panels (b) and (d) present the results of the M-K test of the mean annual $SD_{overall}$
and the mean $SD_{max}$ in China, respectively.
The results of the M-K trend test are the same as those of the slope method (Figure 4). In China, the
overall trends of $SD_{overall}$ and $SD_{max}$ first increase, then decrease, and finally increase again during the
period from 1951 to 2018. The trend of $SD_{overall}$ changes in 1961 and 1997, whereas the $SD_{max}$ trend
changes in 1958 and 2007. Table 3 shows that the break point of $SD_{overall}$ occurs in 2003; for $SD_{max}$, the
break point appears in 2006. In Northeast China, the $SD_{overall}$ break point occurs in 1999, while the $SD_{max}$
break point is observed in 2004. In northern Xinjiang, the $SD_{overall}$ break point appears in 2006, while the
$SD_{max}$ break point occurs in 2000. In the Tibetan Plateau, the $SD_{overall}$ break point is observed in 1957,
and a break point of $SD_{max}$ appears in 2004. In summary, except for the $SD_{overall}$ break point of the Tibetan
Plateau, which appears in 1957, the $SD_{overall}$ and $SD_{max}$ break points of the three snow cover areas are all
in the vicinity of China and are concentrated between 1999 and 2006.
**Table 3.** Break points of snow cover index variations detected by a moving $t$ test.

|  | $SD_{overall}$ | $SD_{max}$ | SCDs | SOD | SED | SDDs |
|---|---|---|---|---|---|---|
| China | 2003 | 2006 | 1999 | 1959, 1964 | 2004 | 2005 |
| Northeast | 1999 | 2004 | 1999 | 2005 | 2001 | 2000 |
| Northern Xinjiang | 2006 | 2000 | 1993, 2002 | 1970 | 1996 | 1999,2001 |
| Tibetan Plateau | 1957 | 2004 | — | 2001 | 2009 | 2005 |

**4.2  SCDs**
The mean annual SCDs gradually increase with increasing latitude and altitude in China from 1951 to
2018 (Figure 5a), and the mean number of SCDs is 34 days per year. The largest SCDs are observed in
Northeast China at 167 days per year. Significant increases in SCDs are concentrated mainly in western
Northeast China (110°E - 128°E, 40°N - 50°N) from 1951 to 2018 (Figure 5b), and the proportion of
meteorological stations is 6.3% in total. The stations with significant decreases are distributed primarily
throughout central China, and the proportion is 9.3%. However, SCDs in the Tibetan Plateau tend to
decrease. Overall, the SCDs tend to increase at a rate of 0.5 days per decade from 1951 to 2018
throughout mainland China (Figure 6a); the SCDs first increase, then decrease and finally increase again,
and the changes appear in 1962 and 1986 (Figure 6b). Nevertheless, there are no significant trends in
SCDs throughout the study period.
In the three snow cover areas, the SCDs in Northeast China and northern Xinjiang both show
increasing trends, especially in Northeast China, where the increase is significant. The Tibetan Plateau
tends to decrease but not significantly, and the trends of the SCDs in Northeast China, northern Xinjiang
and the Tibetan Plateau are 2.3 days, 0.6 days and -0.5 days per decade, respectively (Table 4). The
overall break point of the SCDs in mainland China occurs in 1999. In Northeast China, the break points
of the SCDs occur in 1999. In northern Xinjiang, break points are observed in 1993 and 2002. In contrast,
there is no significant break point in the Tibetan Plateau. Furthermore, the distribution of the SCD trend
exhibits obvious regional differences.

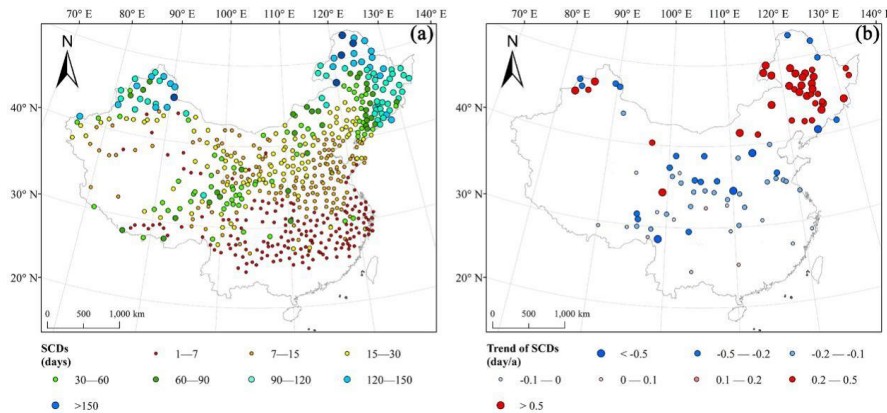

**Figure 5.** Panel (a) presents the spatial distribution of the mean annual SCDs, and panel (b) displays the distribution of the mean annual trend of SCDs with significant changes (*P*< 0.05) as determined by trend analysis.

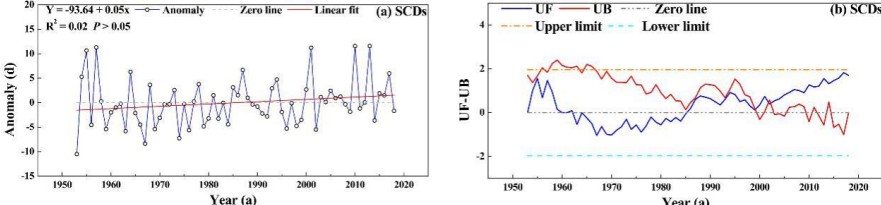

**Figure 6.** Panel (a) is the linear fit of the mean annual SCDs in China. Panel (b) presents the M-K test results for the mean annual SCDs in China.

**Table 4.** Trends in SCDs across the three snow cover areas from 1951 to 2018.

| Zone | Slope analysis | | M-K analysis |
|---|---|---|---|
| | Slope | *P*-value | Z-value |
| Northeast China | 0.23 | 0.01** | 2.46** |
| Northern Xinjiang | 0.06 | 0.33 | 1.02 |
| Tibetan Plateau | -0.05 | 0.16 | -1.41 |

** denotes significance at < 0.01

### 4.3 Snow phenology

Figure 7 shows that the mean SDDs, SODs and SEDs are 99 days, 157th and 256th day, respectively. During the period from 1951 to 2018, the SDDs tend to shorten throughout mainland China, which is caused by the postponement of the SOD and the advance of the SED, and the trends of the SDDs, SOD and SED are -1.4 days, 0.4 days and -0.9 days per decade, respectively. The proportion of meteorological stations with significantly shortened SDDs is 25.2%, while the proportion of meteorological stations with a significant increasing trend in the SDDs is only 0.2%. The stations with significant delays in SOD are concentrated mainly in Northeast China, Northwest China and the Tibetan Plateau (Figure 7d), with a proportion among all meteorological stations of 14.3%. The stations with SODs that have become significantly earlier are mainly concentrated in Southeast China with a proportion of 2.8%. The stations with significant trends in the SED present mainly an advancing trend with a proportion of 17.6%, while the proportion of stations with a significant delay in the SED is only 0.3%. Table 5 shows that in Northeast





China, northern Xinjiang and the Tibetan Plateau, the reductions in SDDs reach 1.9 days, 1.0 days and
4.2 days per decade, respectively, the postponement of SOD is 0.5 days, 0.6 days and 3.3 days per decade,
and the advance of SED is 1.3 days, 0.4 days, and 0.8 days per decade, respectively.
**Table 5** Trends in snow phenology among the three snow areas from 1951 to 2018.

| Zone | Variate | Slope analysis | | M-K analysis |
|------|---------|------|------|------|
| | | Slope | *P*-value | Z-value |
| | SOD | 0.05 | 0.34 | 0.91 |
| Northeast | SED | -0.13 | 0.01** | -2.60** |
| | SDDs | -0.19 | 0.01** | -2.49** |
| | SOD | 0.06 | 0.25 | 1.35 |
| Northern Xinjiang | SED | -0.04 | 0.47 | -1.11 |
| | SDDs | -0.10 | 0.19 | -1.51 |
| | SOD | 0.33 | 0.00** | 4.88** |
| Tibetan Plateau | SED | -0.08 | 0.04* | -2.14* |
| | SDDs | -0.42 | 0.00** | -4.51** |

** denotes a significant change at P < 0.05.

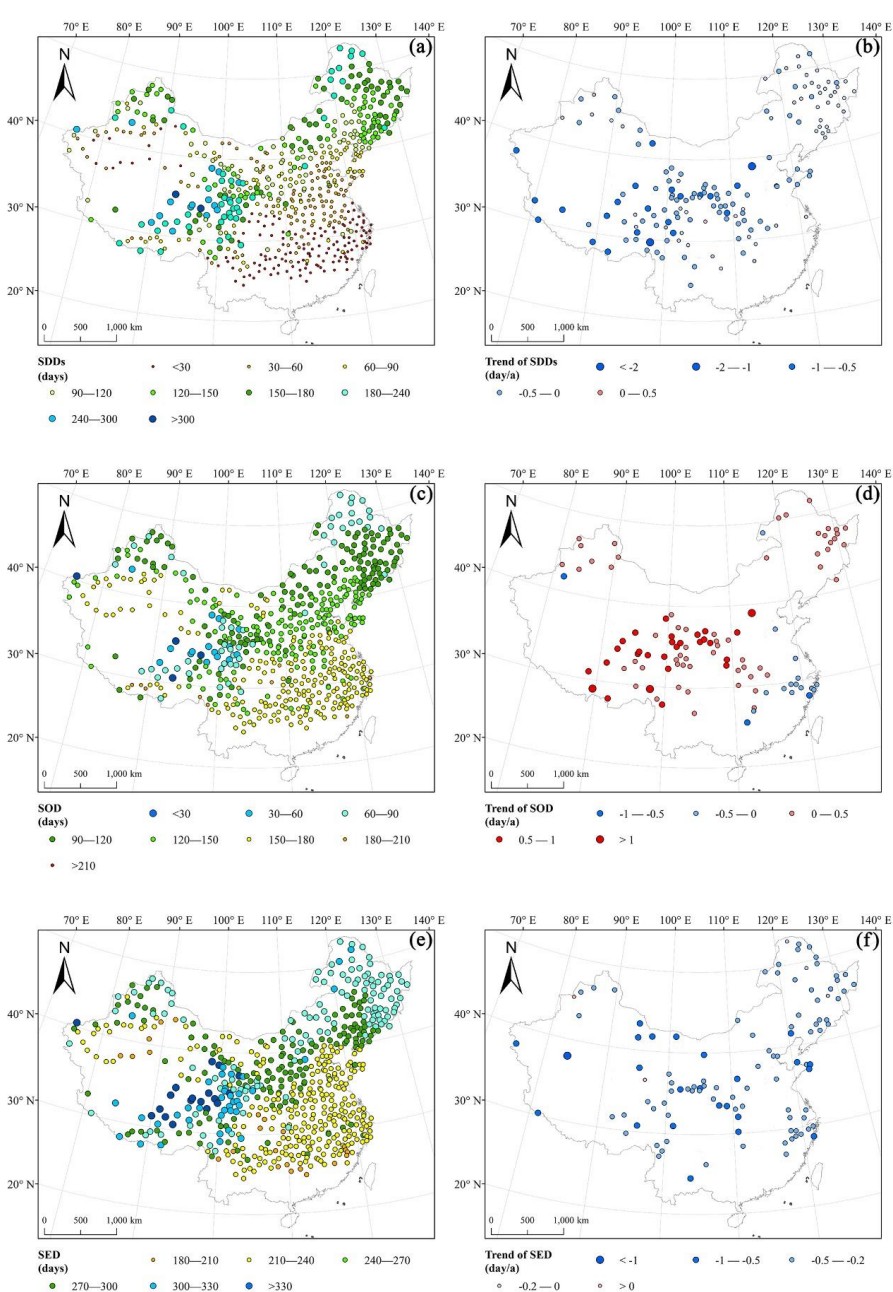

**Figure 7.** Panels (a), (c) and (e) present the spatial distributions of the mean annual SDDs, the mean annual SOD and the mean SED across China, respectively. Panels (b), (d) and (f) display the distributions of the trend of the mean annual SDDs, the mean annual SOD and the mean SED with significant changes ($P < 0.05$) across China as determined by trend analysis.

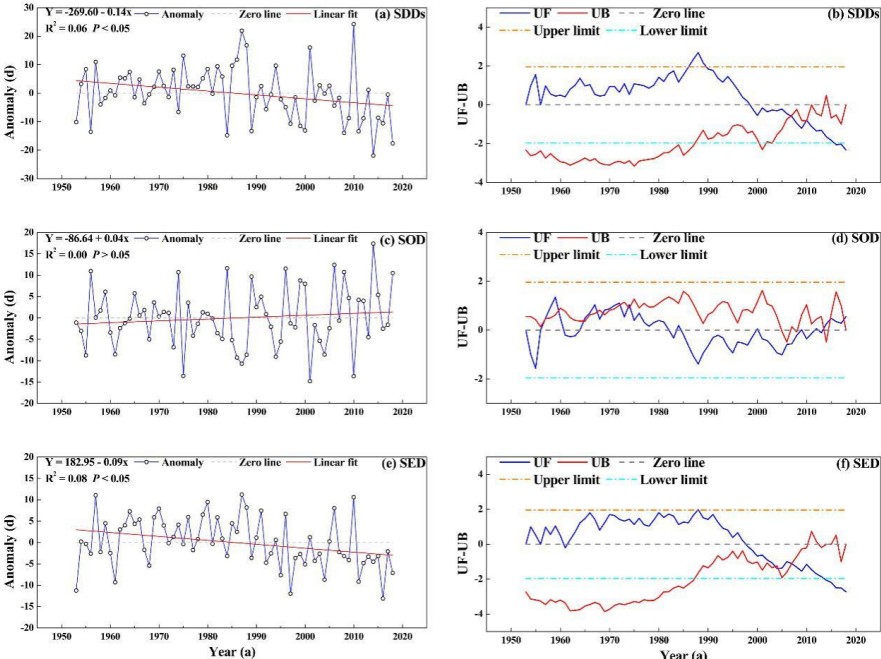

**Figure 8.** Panels (a), (c) and (e) present the linear fits of the mean annual SDDs, the mean annual SOD and the mean SED in China, respectively. Panels (b), (d) and (f) present the M-K test results of the mean annual SDDs, the mean annual SOD and the mean SED in China, respectively.

Figure 8 indicates that the SDDs first increase and then decrease, and the trend transforms in 1999, and significant increases occur from 1987 to 1989. The trend of the SOD changes many times, showing an upward trend from 1964 to 1982, followed by a downward trend from 1985 to 2013. The SED tends to delay at first and then appears earlier, and the trend transforms in 1998. The break point of SDDs occurs in 2005, the SOD break points appear in 1959 and 1964, and the SED break point is observed in 2004. The spatiotemporal variations in snow cover phenology show obvious regional heterogeneity. In Northeast China, the break point of SDDs occurs in 2000, the SOD break point appears in 2005, and the SOD break point is observed in 2001. In northern Xinjiang, the break points of the SDDs are observed in 1999 and 2001, the SOD break point occurs in 1970, and the SED break point appears in 1996. In the Tibetan Plateau, the break point of the SDDs is in 2005, the SOD break point occurs in 2001, and the SED break point appears in 2009. Overall, from 1951 to 2018 in mainland China, the SDDs, SOD and SED are highly related to geographical zonality, and their trends are shortened, delayed and appear earlier, respectively. The main reason for the shortening of SDDs is the SED appearing earlier. However, in the Tibetan Plateau, the main reason for the shortened SDDs is the delay in SOD. Furthermore, the snow phenology changes in the Tibetan Plateau are greater than those elsewhere.

*4.4 Snow heterogeneity*

Figure 9 shows that the trend of annual precipitation shows that the significantly increasing trend is mainly concentrated in western China from 1953 to 2017, especially in Xinjiang and the inner Tibetan Plateau. However, the significantly decreasing trend of precipitation is mainly concentrated in Northeast China. The proportions of the areas with significantly increasing trends and decreasing trends are 20.4% and 6.8%, respectively. The annual mean temperature tends to increase throughout China from 1953 to

2017. In addition, up to 97.2% of the region showed a significantly increasing trend, with a maximum
trend of 0.04 °C per year.

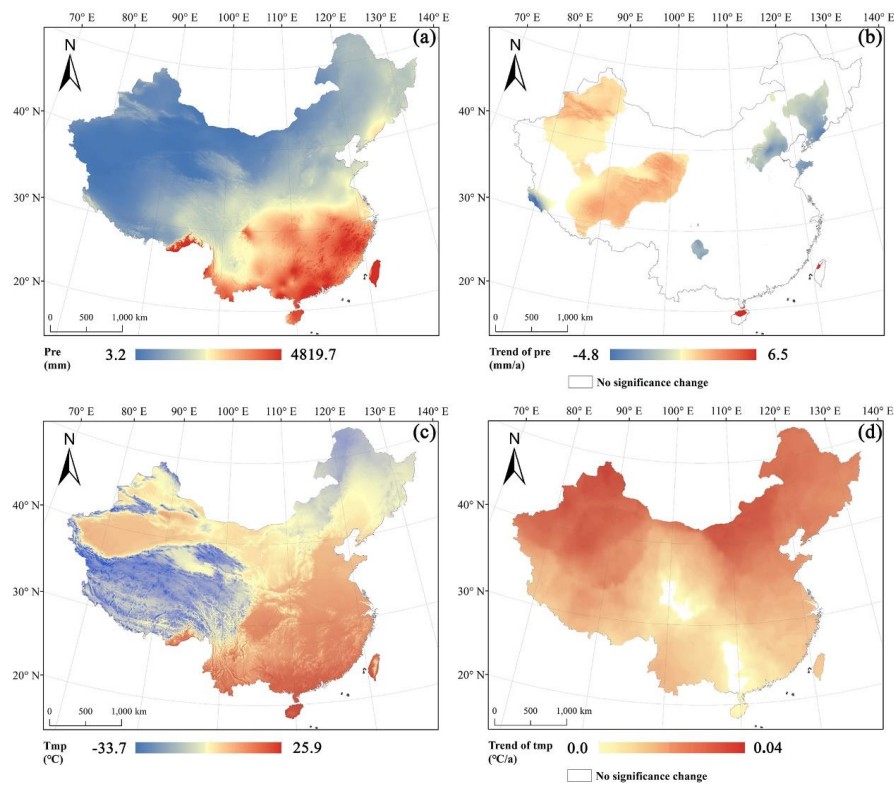

**Figure 9.** Panels (a) and (c) represent the annual mean precipitation and temperature, and panels (b) and
(d) are the significant trends ($P < 0.05$) from 1951-2017 across China.
Figure 10 indicates that the geographical and meteorological factors all affect SD, SCDs and snow
phonology indirectly and directly. Longitude, temperature and precipitation have an impact on $SD_{overall}$,
while latitude and altitude do not impact $SD_{overall}$, of which the maximum total standardized effect is the
annual coldest monthly minimum temperature (Tmn-min), with a coefficient up to 0.9. All factors affect
the spatial and temporal distribution of $SD_{max}$, and interestingly, the annual warmest monthly maximum
temperature (Tmx-max) is the most important factor controlling the $SD_{max}$, and the coefficient of the total
standardized effect is 0.8. This factor was followed by the annual total precipitation (Pre-sum), Tmn-min
and longitude. The SCD distribution is also controlled by all geographical and meteorological factors, of
which the Tmn-min is the most important factor with a total standardized effect of 1.5. The geographical
and meteorological factors have an impact on the SOD, but the altitude is the most sensitive to the start
time of snow cover. The higher the altitude is, the earlier SOD is. The most important factor that controls
SED is the annual mean temperature (Tmp-mean), with the largest total standardized effect coefficient
of 1.1. Altitude is another important factor affecting SED. However, SCDs include only empirical
ephemeral snow days compared to SDDs, and the contributions to SCDs and SDDs are completely
inconsistent. Although Tmn-min is the most important factor controlling the SCD distribution, its effect





on SDDs is negligible. Altitude has the largest total standardized effect coefficient of 0.9, which has the
most important effect on SDDs, followed by Tmp-mean, longitude and latitude.

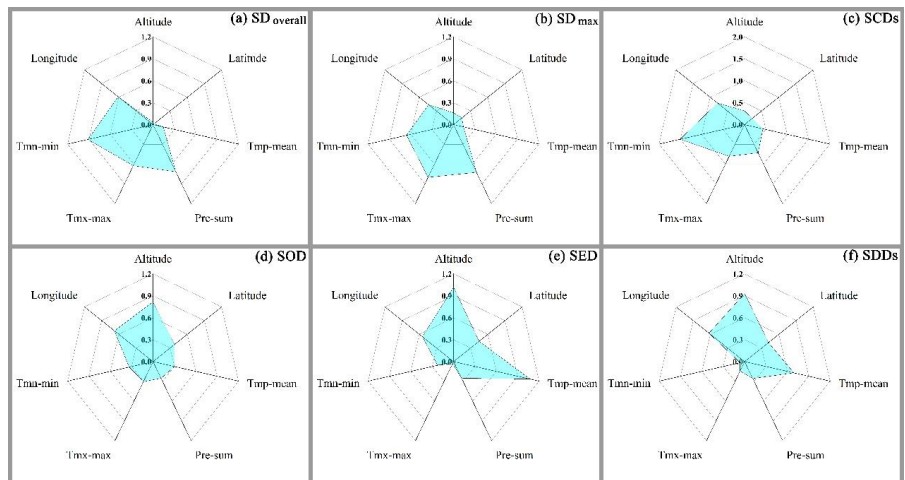

**Figure 10.** Panels (a-g) show the standardized total effects of geographic and meteorological factors on
snow cover parameters, including the $SD_{overall}$, $SD_{max}$, SCDs, SOD, SDDs, and SED.
**5   Discussion**
The variation and distribution of snow cover and its causes have always been a popular topic. In this
study, snow indices calculated based on data from meteorological stations over the past 60 years were
analyzed in China. The overall SCDs in China are increasing, the SOD is delayed, the SED has been
appearing earlier, and the SDDs are shortened. Nevertheless, the SD shows an increasing trend, especially
in Northeast China. The snow parameters show different oscillatory periods also found in this study
between 1951 and 2018.

13        Regarding the SD, using passive microwave remote sensing, Che et al. (2008) found that the SD in

China showed a weak increasing trend from 1978 to 2006. Zhong et al. (2018) found that both the annual
mean and the maximum SD showed increasing trends over the entire Eurasian continent, including China,
and the snow depth decreased in autumn and increased in spring and winter. The results of our study are
roughly the same as those of this previous study. The areas where $SD_{overall}$ and $SD_{max}$ increased
significantly were mainly concentrated at latitudes above 40 °N. The results of both the M-K analysis
and slope methods show similar changes in Northeast China and northern Xinjiang. Huang et al. (2016)
also found a significant increasing trend of SD in Northeast China, whereas SD decreased in the north
and northwest regions of the Tibetan Plateau from 2000 to 2014 according to MODIS snow products.
This study also found decreasing trends of $SD_{overall}$ and $SD_{max}$ from 1951 to 2018 in the Tibetan Plateau.
Wei & Dong (2015) used the results of CMIP5 multimodel averages to assess and simulate the SD in the
Tibetan Plateau, and they found that SD indicated decreases for most of the models from 1851 to 2005.
However, the results from CMIP5 overestimated snow depth over the Tibetan Plateau.

26        The SCDs at the middle and low latitudes of the Northern Hemisphere, including Northeast China,

showed an increasing trend from 2000 to 2015 according to the MODIS snow cover dataset (Huang et
al. 2016; Wang et al. 2018). A large number of studies have also found that the SCDs on the Tibetan
Plateau show a declining trend (Chen et al., 2015; Huang et al., 2017; Qiao et al., 2018). Hori et al. (2017)
found regional differences in the SCD changes among the three stable snow cover areas in China by



using long-term JASMES (JAXA Satellite Monitoring for Environmental Studies) snow cover products. However, our study obtains similar results by using a dataset from meteorological stations. In our study, we found that the SCDs increased in Northeast China, northern Xinjiang and even throughout China. However, the SCDs of the Tibetan Plateau shortened from 1951 to 2018. This result is also similar to the results obtained by Huang et al. (2017), who used MODIS daily snow cover products from 2001 to 2014.

Regarding snow phenology, Ke et al. (2016) found that from 1952 to 2010, the overall snow phenology in China reflected a delay of the SOD and an advancement of the SED. Peng et al. (2017) found that compared to the entire Northern Hemisphere, there was a significant increase in the SEDs that appeared earlier among the three stable snow cover areas in China from 1979 to 2006. This result is similar to the results of our study. The snow phenology indicated a shortened number of SDDs, a delay in the SOD and an advancement in the SED in China; among them, the SDDs and the SED changed significantly from 1951 to 2018. In Northeast China, the trends of SOD, SED and SDDs are similar to those in China. In particular, the SED has been appearing earlier and contributes more to the reduction in SDDs. However, in the Tibetan Plateau, the trends of SOD, SED and SDDs are all significant. The trend of SOD (0.3 days per year) is much larger than that of SED (-0.1 days per year). Therefore, the main reason for the shortening of the SDDs in the Tibetan Plateau may be the significant delay of the SOD. Using a combination of these six snow cover indices, we found that the $SD_{overall}$ and $SD_{max}$ in China increased, especially in Northeast China and northern Xinjiang, from 1951 to 2018. However, the $SD_{overall}$ and $SD_{max}$ on the Tibetan Plateau showed a weakly decreasing trend. In summary, more SCDs and shorter SDDs led to increased snow cover across China from 1951 to 2018.

In mainland China, the distribution of snow cover exhibits strong zonality, and the snow depth and its phenology are highly spatially and temporally heterogeneous. A large number of studies have proven that meteorological and geographic factors are closely related to snow cover (Ye, 2018; Zhong, 2018). However, geographical factors are a direct factor that affects climate and indirectly affects snow cover distribution. Therefore, meteorological factors, including temperature and precipitation, are very important factors affecting snow cover. This study shows that meteorological factors not only affect the depth of snow cover but also have an important influence on the phenology of snow cover; in particular, the influence of the temperature effect on snow is obvious. With climate change, especially climate warming, snow depth and its phenology will change severely, which will dominate the rise in temperature in the future.

## 6 Conclusion

The variation and distribution of snow cover have always been popular research topics. Here, we assessed snow cover variations using in situ observations provided by the CMA across mainland China from 1951 to 2018. The conclusions of this study are as follows:

1) Snow depth tends to increase in China, of which the overall $SD_{overall}$ and $SD_{max}$ tend to increase significantly in Northeast China, and nonsignificantly in northern Xinjiang. However, $SD_{overall}$ and $SD_{max}$ both show decreasing trends in the Tibetan Plateau, but the trend is negligible.

2) SCDs in Northeast China and northern Xinjiang both show increasing trends, especially in Northeast China, where the increase is significant. However, the SCDs in the Tibetan Plateau tend to decrease but not significantly.

3) SDDs tend to shorten throughout mainland China, which is caused by the SOD delay, and the SED has been appearing earlier. Furthermore, the snow phenology changes in the Tibetan Plateau are greater than those elsewhere.



4) Geographical and meteorological factors are the main factors controlling the heterogeneity of the spatial distribution of snow depth and phenology, while changes in meteorological factors have a more important influence on snow cover distribution, especially changes in temperature, which will lead to significant changes in snow depth and phenology.

*Acknowledgments.* The authors acknowledge the snow depth dataset support from the National Meteorological Information Center of the China Meteorological Administration (CMA) (http://data.cma.cn/en). Meteorological data were obtained from the Loess Plateau Data Center, National Earth System Science Data Sharing Infrastructure, National Science & Technology Infrastructure of China (http://loess.geodata.cn).

*Funding.* This research has been supported by the Natural Science Foundation Projects of China (41971293; 41671330; 41871238), the Science and Technology Basic Resource Investigation Program of China (2017FY100501), and the Startup Foundation for Introducing Talent of Nanjing University of Information Science & Technology (20191017).

*Author contribution.* X.D. Huang conceived the research and wrote the paper. T.G. Liang and Z.J. Zheng guided the implementation of the research, and revised and finalized the manuscript. C.Y. Liu provided data analysis and graphic drafting. Y.L. Wang provided the data analysis and suggestions.

*Competing interests.* The authors declare no conflicts of interest.

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
