# Peer review of "Snow cover variations across China from 1951-2018"

_The Cryosphere, 2020_

## Referee Comment (RC1) · Anonymous Referee #1 · 2 Sep 2020

In this paper the authors' document trends in China snow cover from surface observations of daily snow depth over the period 1951-2018. The paper results are consistent with previous publications showing mainly increasing snow depth and snow cover above 40N, with decreasing snow cover south of 40N. The main merit of the paper is the period of record (1951-2018) which currently represents the most up-to-date (and longest) assessment of snow cover trends in China. The introduction is well-written and comprehensive, but would be improved with more focus and synthesis of the Chinese snow cover literature, and a clearer discussion and presentation of the study rationale. The data and methods sections are mostly well written, although the methods section could use some additional explanation in a few places (see detailed comments). The trend results are presented clearly, but there is considerable potential to streamline

the presentation. The Structural equation modeling component of the analysis is not compelling; it currently lacks a clear rationale and is based on inappropriate air temperature and precipitation variables. The updated trend results presented in the paper are of strong interest to the cryospheric community. However, the paper provides little explanation of the mechanisms responsible for the trends, which is a major weakness.

Detailed comments: 1. Suggested wording change for first line of Abstract: "Snow cover changes over China from 1951 to 2018 are documented based on an analysis of in situ daily snow depth observations from 730 meteorological stations. The snow cover indicators analyzed included snow depth (SD), snow covered days (SCDs), and snow phenology."

2. The Introduction is well written and comprehensive, but it needs to focus more on China snow cover. I think you could delete the first two paragraphs and replace this with a focussed discussion of the various advantages and disadvantages of the currently available data for monitoring snow cover changes over China. In this regard, your statement that in situ snow depth observations provide the most reliable dataset for analyzing snow cover changes "with a high degree of credibility" will need to acknowledge the strong local scale processes influencing point snow depths, the poor spatial distribution of stations in some regions of China, and the low-elevation bias in the station network.

A summary table of China snow cover trend results from previous studies would be a useful addition to the Introduction given the sensitivity of trend results to the specific period of data analysed. The recent findings by Ma et al. (2020) of the role of changes in winter snow-free periods in snow cover duration trends should be included in the discussion as they help explain why snow cover duration can increase under warming temperatures. A concise synthesis of previous results and identification of knowledge gaps is important for providing a clearer rationale for this particularly study. For example, the SEM analysis presented in subsection 3.2 appears to be an innovative aspect of the paper that needs to be incorporated in the study rationale.

3. What is your definition of "stable" snow cover? Are TP, NX and NC highlighted in Figure 1 because they are the only areas in China with a stable snow cover? Is this also the reason that only these three regions are summarized in the results? The issue of ephemeral vs stable snow cover deserves some discussion particularly in light of the Ma et al. (2020) paper. Related to this, your statistical analyses are carried out at all stations in China not just those with a stable snow cover. How robust are the statistical methods in ephemeral snow cover areas with frequent zero snow cover years and undefined start and end dates to the snow season?

4. In Section 2.2 the use of an annual period seems strange given the snow season is confined to a much shorter cold season. It is also not clear how annual maxima of air temperature and precipitation would assist in diagnosing changes in snow cover. From energy and mass balance considerations variables like freezing degree-days, total solid precipitation and the solid-fraction of total annual precipitation would be expected to have more relevance for explaining variability and change in snow cover.

5. Section 3.1: Can you provide a line or two of text prior to eqns. 3 to 6 to explain what these equations are being derived for? I suggest you add a new section "3.2 Change-point analysis". Overall I find section 3.1 a bit confusing and statements in the Results section further increase my confusion e.g. page 9 line 4 "The results of the M-K trend test are the same as those of the slope method".

6. Section 3.2: Please provide some introductory text to your current section 3.2 on why you proposing to employ SEM? What are the hypotheses you are testing and why is SEM the best method? In your statement that "seven factors were screened out", I think you mean that seven factors were retained for analysis. As mentioned previously, the use of annual maxima in this analysis is difficult to justify for understanding snow cover variability.

I think you would learn more about snow cover variability by defining the regional snow cover response regions from EOF analysis, then looking at the corresponding regional

time series of snow season air temperature, total precipitation, and precipitation solid fraction.

7. Include slope units in Table 2. Why is the China average not included as in Table 3? The same applies to the trend result tables for other snow cover variables.

8. Can you explain how the anomaly time series in Figure 4 is obtained? There should be an error bar for each annual mean, and the error will influence the linear fit through the points. Can you also provide a brief explanation how to interpret the UF and UB curves in Figures 4b and 4d. Wouldn't the confidence interval in the trend get narrower as the length of the time series increases?

9. What is responsible for the break points shown in Table 3? Are they linked to changes in atmospheric circulation?

10. Section 4.4: The results of the SEM analysis are not very convincing and add little to the paper. The analysis may be more instructive using air temperature and precipitation variables that are more closely linked to snow cover variability.

11. I think your Results section could be significantly shortened if you presented all the snow cover variables together instead of separately. I think this would also help interpreting and explaining the results. At the moment the results are presented in a rather descriptive way following the same format for each variable, which is not very interesting from the readers point of view.

12. The conclusions are largely descriptive and it is hard to find any new insights into snow cover variability and change in China in this paper. As it stands, the only significant contribution of the paper is to extend the period of previous trend analyses. I see several areas where the authors could make potential new contributions: - document the snow response regions of China from EOF analysis of station annual series of SDmax and SCD series - determine the roles of regionally-averaged (over the identified snow response regions) snow season air temperature, total precipitation, and

total snowfall in the observed snow cover series. - find physical explanations for break points e.g. atmospheric circulation changes, increased snowfall in winter storms, fewer snow-free periods (e.g. Ma et al. 2020). - find physical mechanisms for the decrease in snow season gaps documented by Ma et al. (2020)

Literature cited: Ma, N., Yu, K., Zhang, Y., Zhai, J., Zhang, Y. and Zhang, H., 2020. Ground observed climatology and trend in snow cover phenology across China with consideration of snow-free breaks. Climate Dynamics, pp.1-21.
* * *

---

## Referee Comment (RC2) · Anonymous Referee #2 · 6 Sep 2020

General comments: The author gives us impressive work on snow cover variation analysis using more than 60 years meteorological station observation. Huang et al. investigated the snow cover variation characteristics with SD, SCD and snow phenology, and provide detailed spatial and temporal characteristics of snow cover in China since from 1951 to 2018. This manuscript gives a contribution to understanding the snow cover variation in China. Although M-K test which gives a break point indicating snow cover variation trends is interesting method, the authors pay little attentions to conclude and introduce this part work. This manuscript, however needs revision with regards to the organization and presentation of the results.

Major Comments 1: As described in introduction "The results indicated that from 1952 to 2010, the overall snow phenology in China reflected a delay of the snow onset date

(SOD) and an advancement of the snow end date (SED).(Ke et al., 2016)" (page 3 lines 6-7) this is similar with your finding except the dataset used in this study is longer than Ke. My first question. What's the different or new finding derived from your analysis results when using the 1951-2018 stations data?

2: M-K test results, which I think it is very good for understanding snow cover variation in China, need more efforts to enhance presentation in Abstract, Conclusion section. As suggested by Reviewer 1#, it needs to give more explanation on M-K test results.

3: Previous studies found snow cover have significant changes occurring at around 2000. And also, this study using M-K test show the break point is after 2000. You just showed this break point results and do not provide further analysis. Previous study by remote sensing snow depth data has investigated the different variation rate between before and after 2000 (Xiao et al. 2020 and other studies). Further analysis is not mandatory. This suggestion is for reference only.

4: Except that Reviewer 1# suggested potential contribution aspects (point 12), in Xiao et al. (2020) study, he found different variation trends for snow depth and snow cover days in some area of Norther Hemisphere (including China), inverse trend or same trends. One potential contribution idea is that linking the variation trends of different indexes to find different response on climate change background. Xiao et al. (2020) study may give you instruction to exhibit insight variation analysis results of snow cover indexes (SD, SCD ...) from 1951-2018.

5. Actually, the threshold selection of snow depth has effect on SCD or SOD or SED or SDDs variation analysis (Dyer et al., 2006; Notarnicola 2020). In previous studies, many kinds of threshold have been applied to define snow-covered and snow-free, e.g., 0cm, 1 cm, 2cm, 5 cm, 10 cm. In the discussion section, this should be added in your analysis and discussion.

Specific comments: Abstract: 1)Remove "retrieved" in page 1 line 12. Snow depth is measured in each meteorological station, not be retrieved.

TCD
2)This term "snow phenology" is not familiar with most of readers. Give a short definition,

3)Change "higher than 40°N" to "northward of 40°N" in line 17

4) "This result was mainly caused by the postponement of the snow onset date and the advancement of the snow end date." Please rephrase this statement. As for the reason of the decrease of snow cover duration, it always should be related to precipitation or air temperature or atmospheric circulation, polar sea ice etc.

5)Please add the more description of M-K test results. I think this is very interesting method for snow cover variation analysis and give a novel finding that the break point of snow cover variation is after 2000.

Introduction 6)Line 28. Please add a reference for the specific number of snow cover area.

7)Line 40. Change "the dataset is" to "this dataset is"

8)What's the meaning of "the statistical significance of the linear trend is very weak"?

9)In introduction section, you give more literature review son snow cover area. But, your study does not give snow cover area variation analysis. Recommend to only give short description on this topic.

10)"Poor accuracy"? I don't think so. "short observation periods"?? According to your introduction, the NOAA snow cover extent data provide a long-term snow dataset (1967-present) "Snow Lab and the binary snow cover mask data derived from the Climate Data Record of the Northern Hemisphere Snow Cover Extent (NHSCE) can provide a long-term snow dataset (1967-present)." In page 1 line 38-39. Please rephrase this sentence.

11)In this section, you gave so many literature reviews on remote sensing snow cover monitor results, but little on stational observation results. Please reorganize this section

TCD
statement. Recommend to emphases the stational snow cover analysis results

Dataset 12)The caption of Figure 1. Please give description on the numbers in parenthesis

13) "Snow depth of ground observations is measured manually with a wooden ruler at 8 o'clock every day when the ground in the field of view around the meteorological station is covered by more than half in snow." It's a valuable information for understanding snow depth measurement at meteoritical station. Please add a reference.

14)Page 4 line 19. Remove "...from the analysis in this study."

Methodology

15) Change the title of Section 2 "dataset and methodology" to "Dataset". The section 3 title is "Methodology"

16) Page 5 line 16. "in the snow cover indices from 1952 to 2012", is it should be "1951 to 2018"?

17) Page 6 line 2. What's meaning of "UB"

18) Page 6 line 3-11. You just list a series of formulas. Actually, I don't understand what's UB and UF stand for. Recommend to add more introduction information for Eq. 3 - Eq. 6.

19) Page 6 line 27-29. "... assume that latitude, longitude and altitude directly affect precipitation and temperature and thus indirectly affect snow cover, while precipitation and temperature have a direct impact on snow cover" Please give other publications to support.

20) Which threshold was used in this study to transform snow depth to snow-covered or snow-free?

Results
21) Page 7 line 6: "mean annual SD"? But "annual mean SD" was used in above section. Please modified.

22) The legend in Figure 3b. why did not use "< -0.1; -0.05 $\sim$ -0.1; -0.05 $\sim$ 0; 0 $\sim$ 0.05; 0.05 $\sim$ 0.1; > 0.1"?

23) Section 4.1. page 9 lines 4-13. I think that the results of the M-K trend test (Table 3) are very valuable presentation and it give a great contribution to the snow cover variation study/research. You just offer descriptive information. I suggest that you should provide further explanations to analyze these results. What changes in climate could contribute to this break point. I am looking forward to your further analysis results in this part.

24) Similar comments to Section 4.3 (page 13 lines 5-19)

25) Section 4.3 "157th and 256th". Please give start time (1st January or 1st September) and add the specific time for these two dates, for example 7th (7th January)

26) As suggested by Reviewer 1#, the Result section could be shortened. It's helpful to put more attention on snow cover variation results analysis and the new finding interpretation. From Table 3, I find that almost all indexes (SD, SCD, SOD, SED and SDD) break point occur in the new century (after 2000s). You can give further analysis on what's the different variation rate before and after break point for these indexes.

27) Section 4.4 in line 21 page 13. change "annual precipitation" to "annual mean precipitation"

28) Page 14 lines 7-12: (Q1): why did the latitude and altitude have different effects on SD\_overall and SD\_max? "latitude and altitude do no impact SDoverall" but "all factors affect the spatial and temporal distribution of SDmax". (Q2): according to previous studies, latitude and altitude have a significant effect on SD. Your conclusion is "latitude and altitude do not impact SDoverall". Please give me more explanations.

29) As we all known, MODIS do not provide SD information. In page 15 line 19-21, "...
whereas SD decreased in the north and northwest regions of the Tibetan Plateau from 200 to 2014 according to MODIS snow products". Please revised it. Especially, Figure 3 do not have significant change station in northwest of the Tibetan Plateau!

Reference: Xiao, X.; Zhang, T.; Zhong, X.; Li, X. Spatiotemporal Variation of Snow Depth in the Northern Hemisphere from 1992 to 2016. Remote Sens. 2020, 12, 2728.

Notarnicola C., Hotspots of snow cover changes in global mountain regions over 2000–2018. Remote sensing of Environment, 2020, 243, 111781

Dyer, J. L. and Mote, T.: Spatial variability and trends in observed snow depth over North America, Geophys. Res. Lett., 33, L16503, 2006.

---

## Author Comment (AC2) · 1 Nov 2020

Anonymous Referee #2 General comments: The author gives us impressive work on snow cover variation analysis using more than 60 years meteorological station observation. Huang et al. investigated the snow cover variation characteristics with SD, SCD and snow phenology, and provide detailed spatial and temporal characteristics of snow cover in China since from 1951 to 2018. This manuscript gives a contribution to understanding the snow cover variation in China. Although M-K test which gives a break point indicating snow cover variation trends is interesting method, the authors pay little attentions to conclude and introduce this part work. This manuscript, however needs

revision with regards to the organization and presentation of the results. Author's response: Firstly, on behalf of all authors, we appreciate your careful review and also great comments for this manuscript. Please accept our respect and gratitude to you for your pertinent suggestion and responsible review. Base on your comments, the revised manuscript has made the following changes: 1) The Introduction section was revised based on your comments. Currently various available data for monitoring snow cover observations are referred to, including their advantages and limitation. More literatures focus on snow cover variation in China are cited and discussed. And what issues of snow cover change in China still need to resolve was put forward. 2) We have added more principles description of methods used in the article. Only the climate data in the cold season was re-analyzed in the revised manuscript. We definitely found more interesting results this time. 3) The results of the breakpoint analysis were discussed separately. 4) The structure of the article was re-organized, the results were partially condensed and more discussion has been added in order to explain the mechanisms responsible for the trends of snow cover in China. Include an enhance correlation analysis between climate and snow cover, snow cover spatiotemporal pattern based on EOFs analysis, as well as the oscillation cycle based on Morlet wavelet.

Major Comments: 1. Comments from Referees: As described in introduction "The results indicated that from 1952 to 2010, the overall snow phenology in China reflected a delay of the snow onset date (SOD) and an advancement of the snow end date (SED).(Ke et al., 2016)" (page 3 lines 6-7 ) this is similar with your finding except the dataset used in this study is longer than Ke. My first question. What's the different or new finding derived from your analysis results when using the 1951-2018 stations data? Author's Response: Thank you for the comments. Ke et al. (2016) documented the SCDs and snow penology variation, and the relationship the SCDs with temperature and AO index in China from 1952 to 2010. The overall trend of the SCDs, and snow phenology in the two papers was basically consistent, but the dispute was in the northeast of China. According to the observation results of the stations in this region, both the snow depth and SCDs tended to increase, while the Ke's paper showed that

the SCDs was decreasing. Secondly, Ke's paper emphasizes the abnormal year analysis as well as the correlation with the temperature and AO. Our paper is more focus on the snow cover trend and driving factors in the long time series, and the study finds that warming climate in the clod season is the primary driving factor for the variation of snow cover. In addition, from the perspective of climate change, the warming trend of China has slowed down after 2000, in our study, we found that snow cover has undergone a abrupt change since 2000, which is also the biggest finding of this paper.

2. Comments from Referees: M-K test results, which I think it is very good for understanding snow cover variation in China, need more efforts to enhance presentation in Abstract, Conclusion section. As suggested by Reviewer 1#, it needs to give more explanation on M-K test results. Author's response: Revised as you suggested. The results from M-K were summarized in Abstract and Conclusion section. And more explanation was added to those equations belong to M-K analysis, and a new section for the breakpoint test using M-K was added. Thank you.

3. Comments from Referees: Previous studies found snow cover have significant changes occurring at around 2000. And also, this study using M-K test show the break point is after 2000. You just showed this break point results and do not provide further analysis. Previous study by remote sensing snow depth data has investigated the different variation rate between before and after 2000 (Xiao et al. 2020 and other studies). Further analysis is not mandatory. This suggestion is for reference only. Author's response: Thank you for the comments. In the revised manuscript, results from M-K were analyzed in detail, not only the snow cover variation but also the temperature and precipitation. Results indicated that the annual mean air temperature in the cold season experienced several oscillations before 1980s, but begun warming persistently after 1980s, and warmed significantly after 1990s, then slowed down after 2000s during the period of 1951 to 2018 in China. Interestingly, the years with positive anomaly of snow depth corresponded perfect with negative anomaly of temperature and negative anomaly of precipitation. Although both temperature and snow

depth showed a significant increasing trend, they were not positively correlated, but significantly negatively correlated which tested by Pearson correlation analysis. The study found that the abrupt change of snow cover occurred mainly around 2000. What causes the abrupt change of snow cover needs to be further explained in combination with extreme weather, atmospheric circulation, etc.

4. Comments from Referees: Except that Reviewer 1# suggested potential contribution aspects (point 12), in Xiao et al. (2020) study, he found different variation trends for snow depth and snow cover days in some area of Norther Hemisphere (including China), inverse trend or same trends. One potential contribution idea is that linking the variation trends of different indexes to find different response on climate change background. Xiao et al. (2020) study may give you instruction to exhibit insight variation analysis results of snow cover indexes (SD, SCD : : :) from 1951-2018. Author's response: Thank you for the comments. In the revised manuscript, an enhance correlation analysis between climate and snow cover, snow cover spatiotemporal pattern based on EOFs analysis, as well as the oscillation cycle based on Morlet wavelet. Our results indicated that the large-scale fluctuation of snow cover must be the result of climate change, and the abnormal events mainly due to the short extreme weather. This study also found that the temperature effect on snow cover is more important than precipitation obviously, and the snow cover varies interannual, the snow depth fluctuates, and changes periodically with the increase of temperature. With climate change, especially climate warming, snow depth, and its phenology will change severely, which will dominate by the temperature rise in the future.

5. Comments from Referees: Actually, the threshold selection of snow depth has effect on SCD or SOD or SED or SDDs variation analysis (Dyer et al., 2006; Notarnicola 2020). In previous studies, many kinds of threshold have been applied to define snow-covered and snow-free, e.g., 0cm, 1 cm, 2cm, 5 cm, 10 cm. In the discussion section, this should be added in your analysis and discussion. Author's response: We agree with the comments that the threshold selection of snow depth has effect on snow indices variation analysis. Due to the large spatial difference of snow depth in China, especially in Tibetan Plateau, the snow depth is generally shallow (average annual snow depth is less than 5 cm). According to the snow cover observation specifications of stations, in order to effectively compare all stations under a unified framework, a threshold greater than and equal to 1cm was used in this study when calculate all the snow cover indices. The threshold selection issue may effect on snow variation analysis was added in discussion section. Thank you very much.

Specific comments: Abstract: 1. Comments from Referees: Remove "retrieved" in page 1 line 12. Snow depth is measured in each meteorological station, not be retrieved. Author's response: Changed as you suggested. Thank you. Author's changes in manuscript: "Snow cover changes over China from 1951 to 2018 are documented based on an analysis of in situ daily snow depth observations from 730 meteorological stations. The snow cover indicators analyzed included snow depth (SD), snow covered days (SCDs), and snow phenology."

2. Comments from Referees: This term "snow phenology" is not familiar with most of readers. Give a short definition. Author's response: Defined as you suggested. Thank you. Author's changes in manuscript: "And the snow duration, onset and end dates in a given year, which are collectively referred to as snow phenology, are becoming increasingly valuable indicators of climate change"

3. Comments from Referees: Change "higher than 40_N" to "northward of 40_N" in line 17. Author's response: Changed as you suggested. Thank you. Author's changes in manuscript: "The significant increases were concentrated at latitudes northward of 40°N, especially in Northeast China." 4. Comments from Referees: "This result was mainly caused by the postponement of the snow onset date and the advancement of the snow end date." Please rephrase this statement. As for the reason of the decrease of snow cover duration, it always should be related to precipitation or air temperature or atmospheric circulation, polar sea ice etc. Author's response: Revised as you suggested. Thank you. Author's changes in manuscript: "Regarding the snow phenology

variations, the snow season in China shortened, and 25.2% of the meteorological stations showed significant decreasing trends, jointly resulting from later starting dates and earlier ending dates. The change of snow phenology is mainly caused by warming in the cold season in China." 5. Comments from Referees: Please add the more description of M-K test results. I think this is very interesting method for snow cover variation analysis and give a novel finding that the break point of snow cover variation is after 2000. Author's response: Did as you suggested. Thank you. Author's changes in manuscript: The M-K test results are summarized as following: "Figure 6 shows the trend of different snow cover indexes changing with time during the period from 1951 to 2018 in China. From the perspective of time variation trend, each snow cover index has its own characteristics over time, which does not change linearly, but shows an increasing or decreasing trend of fluctuation and oscillation. The overall trend of snow depth is increasing, but the change has roughly experienced the process of first increasing, then decreasing, and then increasing again. The trend of SDoverall changes in 1961 and 1997, and increase significantly after 2010. The trend of SDmax changes in 1958 and 2007, however, the overall increase was not significant. SDDs represents the length of the snow season, while SCDs represents the number of days the surface is covered by snow. During the period of 1951 to 2018, the SCDs has experienced three processes of increasing first, then decreasing, and then increasing, with the overall increasing trend but not significant. While the SDDs first increase before 1999 and then decrease, and reduction significantly after 2015, indicating that the shortening of snow season in recent years is a well-established fact. The change of SOD is relatively complex. The SOD experienced several trends of postponing or advancing, and the overall trend was delayed but not significant. SED showed a trend of delay until 1998, but then getting earlier, and advance significantly after 2014. Table 3 summarized the breakpoints of six snow cover indices detected by a moving t test. The results indicate that the spatiotemporal variations in snow cover show obvious regional differences. Specially, there is no significant breakpoint in the Tibetan Plateau. Furthermore, the snow phenology changes in the Tibetan Plateau are greater than those

elsewhere. The abrupt change years of snow cover in different regions were almost different. However, almost all the abrupt change occurred around the 2000s, indicating that snow cover had changed significantly in the 20th century in China since 1951." Introduction 6. Comments from Referees: Line 28. Please add a reference for the specific number of snow cover area. Author's response: The introduction section almost rewrote in the revised manuscript. More literatures focus on snow cover variation in China are cited and discussed in introduction section. Thank you for the comments.

7. Comments from Referees: Line 40. Change "the dataset is" to "this dataset is" Author's response: Changed as you suggested. Thank you. Author's changes in manuscript: "However, this dataset is appropriate for only large-scale snow extent studies because of the coarse spatial resolution (24 km), much coarser for the 1967-1998 portion of the record (190.5 km resolution) (Brown & Robinson, 2011)."

8. Comments from Referees: What's the meaning of "the statistical significance of the linear trend is very weak"? Author's response: The meaning of the sentence is the snow cover extent in winter has on trend with significantly. In the revised manuscript, the review of snow cover extent change in Northern Hemisphere was deleted, and paid more focus on China snow. Thank you very much.

9. Comments from Referees: In introduction section, you give more literature review son snow cover area. But, your study does not give snow cover area variation analysis. Recommend to only give short description on this topic. Author's response: Accepted with your suggested. The introduction section in the revised manuscript was more focus on reviewing the snow depth, and snow phenology variation. Thank you. Author's changes in manuscript: "Snow cover represents an essential component of the energy exchange process and hydrological cycle within the global climate system (Euskirchen et al., 2007; Yao et al., 2013). Snow cover has a unique physical attribute of high albedo (Xiao and Che, 2016), which has a positive feedback effect on climate (Tedesco and Miller, 2007). Within the global hydrological cycle, snow cover not only affects the water cycle but also constitutes a highly crucial form of water storage (Ambadan, 2017; Shams et al., 2018). However, snow can also have negative impacts on human life because snowfall and meltwater are direct causes of snowmelt erosion, snowmelt floods, avalanches, and other natural disasters (Li & Simonovic, 2010; Chen et al., 2016). Currently, various snow datasets can be used to evaluate snow cover variations. The Rutgers University Global Snow Lab and the binary snow cover mask data derived from the Climate Data Record of the Northern Hemisphere Snow Cover Extent (NHSCE) can provide a long-term snow dataset (1967-present). However, this dataset is appropriate for only large-scale snow extent studies because of the coarse spatial resolution (24 km), much coarser for the 1967-1998 portion of the record (190.5 km resolution) (Brown & Robinson, 2011). In addition, the implementation of the interactive multi-sensor snow and ice mapping system (IMS) provides another approach for the dynamic monitoring of snow extent (Sönmez et al., 2014). Other sensors with moderate resolution, such as Moderate Resolution Imaging Spectroradiometer (MODIS), can provide global snow extent products with high resolution and accuracy, but the record period is short (2000 to present) (Hall et al., 2002). Passive microwave remote sensing has been regarded as an efficient way to retrieve snow depth (SD) or snow water equivalent (SWE) at hemisphere and global scales, such as the Scanning Multichannel Microwave Radiometer (SMMR), Special Sensor Microwave/Imager (SSM/I), and Advanced Microwave Scanning Radiometer-EOS (AMSR-E). Another technique that assimilates in situ snow depth observations with microwave emissions was applied in the European Space Agency's (ESA) GlobSnow project to estimate the daily SWE time series from 1979 to present over the Northern Hemisphere (GlobSnow v3.0 SWE CDR released in 2019), and this technique is considered to overcome the large errors that rely solely on passive microwave observations (Pulliainen et al., 2006; 2020; Takala et al., 2011). However, while the GlobSnow SWE algorithm exhibits improved accuracy, the data gaps in alpine areas limit its comprehensive use in snow variation assessments. In recent years, the variations in snow cover over China have attracted much attention, especially with regard to the so-called 'third pole' of the Tibetan Plateau due to it being the region with the highest elevation and deepest snow depth at middle latitudes in the Northern Hemisphere (Ma et al., 2010). Early satellite observations in the Northern Hemisphere suggested a decline in the snow cover extent (SCE) over the past several decades, with significantly decrease in spring, and arguably, there has been a slight increase or decrease in winter in satellite observations since the late 1980s in the Northern Hemisphere (Brown and Robinson, 2011; Choi et al., 2010; Cohen et al., 2014; Connolly et al., 2019; Mudryk et al., 2020). The pattern of the snow variations in mainland China is quite different from the Northern Hemisphere observations. The annual mean SCE accounts for 27% of the country's total area in winter, in which the remote sensing observations imply that the average annual SCE decreased during winter and summer but increased in spring and fall from 2000 to 2014, however, these trends were not statistically significant (Huang et al., 2016). Driven by decreased temperature and increased precipitation in the snow accumulation season, the snow cover fraction over mainland China showed an increasing trend of 0.29% decade-1 during 1982–2013, which was significant at the 0.05 level (Chen et al., 2016). The warming surface temperature has slowed since approximately the 2000s in China (Zhou et al. 2017), and the snow cover change more remarkable after the 2000s correspondingly (Zhang et al., 2020). However, different regions within China often show different climatic trends (Soon et al. 2018). The response of long-term snow cover variability in China to climate warming trend remains unclear partly in reginal and periodical issues. Limited by the coarse spatial resolution from passive microwave remote sensing data and severely cloud obscured from optical remote sensing data, in situ snow observations provide the most reliable dataset for analyzing the changes in snow cover with a high degree of credibility. Moreover, snow parameters are calculated from meteorological station data, which have great advantages in the process of long time series research. However, in situ observations from climate station is insufficient for representing at a regional scale due to spatial discontinuities, irregularities and inhomogeneities in the distribution. Most stations distributed in the flat and open area, and rare distribution in west China, especially the Tibetan Plateau. The stations are lacking in inner and mountain areas in plateau, greatly limits the observation integrity
and representation of space. Nevertheless, the sparse station network still provides the long-term of high-quality ground observation data. Snow depth (SD) is a basic and important parameter of snow cover that plays an important role in hydrological applications, numerical weather predictions, climate change research and land surface process simulations. And the snow duration, onset and end dates in a given year, which are collectively referred to as snow phenology, are becoming increasingly valuable indicators of climate change (Ke et al., 2016; Liston et al., 2011). The previous studies analyzed the spatiotemporal variations of the snow cover, including snow cover extent, snow depth and phenology in China based on various observational data in different periods, and explained the causes of the snow cover variation from the perspective of climate change. However, the distribution and variation of snow cover in China is more complicated because of its vast territory, complex topography and diverse climate types. In addition, under the context of warming climate, is the variation of snow depth in China spatially consistent? Dose the snow cover change in regionally and periodically? Is there breakpoint exist for snow cover variation like climate change? The reliable quantitative knowledge of long-term seasonal snow cover and its trend is still lacking. Therefore, we aim to explore the snow cover variations and its pattern in temporal and spatial, as well as the possible cause based on long-term in situ snow depth dataset from 1951 to 2018 retrieved from meteorological stations in China. "

10. Comments from Referees: "Poor accuracy"? I don't think so. "short observation periods"?? According to your introduction, the NOAA snow cover extent data provide a long-term snow dataset (1967-present) "Snow Lab and the binary snow cover mask data derived from the Climate Data Record of the Northern Hemisphere Snow Cover Extent (NHSCE) can provide a long-term snow dataset (1967-present)." In page 1 line 38-39. Please rephrase this sentence. Author's response: Changed as you suggested. Thank you. Author's changes in manuscript: "Limited by the coarse spatial resolution from passive microwave remote sensing data and severely cloud obscured from optical remote sensing data, in situ snow observations provide the most reliable dataset for analyzing the changes in snow cover with a high degree of credibility. Moreover, snow

parameters are calculated from meteorological station data, which have great advantages in the process of long time series research. However, in situ observations from climate station is insufficient for representing at a regional scale due to spatial discontinuities, irregularities and inhomogeneities in the distribution. Most stations distributed in the flat and open area, and rare distribution in west China, especially the Tibetan Plateau. The stations are lacking in inner and mountain areas in plateau, greatly limits the observation integrity and representation of space. Nevertheless, the sparse station network still provides the long-term of high-quality ground observation data." 11. Comments from Referees: In this section, you gave so many literature reviews on remote sensing snow cover monitor results, but little on stational observation results. Please reorganize this section statement. Recommend to emphases the stational snow cover analysis results Author's response: Thank you for the comments. In the revised paper, we have added a section in Discussion, devoted to the analysis of the ground observations from different literature, and compared with the results in this study. In order to avoid duplication, we deleted the review of previous stational observation results. Thank you.

Dataset 12. Comments from Referees: The caption of Figure 1. Please give description on the numbers in parenthesis. Author's response: Did as you suggested. And Thank you. Author's changes in manuscript: "Figure 1. Geographical locations of meteorological station (proportion of valid yearly records in parentheses) in mainland China. The abbreviations of snow cover areas represent the Tibetan Plateau (TP), northern Xinjiang (NX), and Northeast China (NC)."

13. Comments from Referees: "Snow depth of ground observations is measured manually with a wooden ruler at 8 o'clock every day when the ground in the field of view around the meteorological station is covered by more than half in snow." It's a valuable information for understanding snow depth measurement at meteoritical station. Please add a reference. Author's response: A reference as added you suggested. Thank you. Author's changes in manuscript: "China Meteorological Administration. Specifications

for surface meteorological observations. China Meteorological Press, Beijing, 61-63, 2003."

14. Comments from Referees: Page 4 line 19. Remove "...from the analysis in this study." Author's response: Changed as you suggested. Thank you. Author's changes in manuscript: "Stations with records spanning less than 10 years were omitted to ensure the reasonableness of the statistical analysis."

Methodology 15. Comments from Referees: Change the title of Section 2 "dataset and methodology" to "Dataset". The section 3 title is "Methodology" Author's response: Changed as you suggested. Thank you very much.

16. Comments from Referees: Page 5 line 16. "in the snow cover indices from 1952 to 2012", is it should be "1951 to 2018"? Author's response: Changed as you suggested. Thank you.

17. Comments from Referees: Page 6 line 2. What's meaning of "UB". Author's response: Changed as you suggested. The meaning of UB is the invert sequence of UF. The explanation of UF and UB were described in the revised paper. Thank you. Author's changes in manuscript: "M-K test can not only check the variation overall trend of the sequence but also specify the breakpoint starts. First, for a time series x with n samples, a sequential column Sk was constructed. Under the assumption of random independence of time series, a sequence of statistic UF was defined. Then, in reverse order according to time series x, the above process was repeated to calculate UB."

18. Comments from Referees: Page 6 line 3-11. You just list a series of formulas. Actually, I don't understand what's UB and UF stand for. Recommend to add more introduction information for Eq. 3 – Eq. 6. Author's response: More introduction information for equation was added. Thank you. Author's changes in manuscript: "where Sk is the cumulative count of the number of values at the time i great than at time j. E(Sk) and var(Sk) are mean and variance values of Sk. UF is the standardized value of S, while UB is obtained by inverting the sequence of UF. "

[Figure]

19. Comments from Referees: Page 6 line 27-29. "...assume that latitude, longitude and altitude directly affect precipitation and temperature and thus indirectly affect snow cover, while precipitation and temperature have a direct impact on snow cover" Please give other publications to support. Author's response: The SEM analysis was definitely added little to the paper. After discussion, the authors decided to delete the part of the SEM analysis results. Thank you very much for the comment.

20. Comments from Referees: Which threshold was used in this study to transform snow depth to snow-covered or snow-free? Author's response: Daily snow depth larger than 1 cm were recorded as snow cover; stations with snow depth less than 1 cm were regarded as snow free. The threshold selection due to the large spatial difference of snow depth in China, especially in Tibetan Plateau, the snow depth is generally shallow (average annual snow depth is less than 5 cm). According to the snow cover observation specifications of stations, in order to effectively compare all stations under a unified framework, a threshold greater than and equal to 1cm was used in this study when calculate all the snow cover indices. Thank you for the comment.

Results 21. Comments from Referees: Page 7 line 6: "mean annual SD"? But "annual mean SD" was used in above section. Please modified. Author's response: Modified as you suggested. Annual mean SD were used throughout the revised manuscript. Thank you.

22. Comments from Referees: The legend in Figure 3b. why did not use "< -0.1; -0.05_-0.1; -0.05_0; 0_0.05; 0.05_0.1; > 0.1"? Author's response: Revised based on your suggestion. Thank you. Author's changes in manuscript:

23. Comments from Referees: Section 4.1. page 9 lines 4-13. I think that the results of the M-K trend test (Table 3) are very valuable presentation and it give a great contribution to the snow cover variation study/research. You just offer descriptive information. I suggest that you should provide further explanations to analyze these results. What changes in climate could contribute to this break point. I am looking forward to your

further analysis results in this part. Author's response: Thank you for the comments. In the revised paper, we made great efforts on climate change contributing to snow cover variation. Our study found that the long-term trend of snow cover variation is mainly due to warming climate in the cold season, and the short-term anomaly of snow cover is related to extreme weather, such as the abnormal increase of snow depth in negative temperature anomaly. The abrupt change of snow cover occurred mainly around 2000, but the temperature warming gradually slowed down after 2000 in China. We cannot simply conclude that the abrupt change in snow cover around 2000 mainly due to the climate warming slowed down. Cause the distribution and variation of snow cover in China is highly heterogenetic, which exhibits obvious regional differences. Mainland China with vast territory, complex topography and diverse climate types. The abrupt change in snow cover should be further start with the reasons that affect the reginal water and heat balance of climate system, especially after 2000. We believe that the main contribution of this paper is to find out the existence of these scientific questions, but the physical mechanism needs further analysis on more related to climate system , such as if the atmospheric circulation has changed after 2000, and if the extreme weather has increased after 2000, etc.

24. Comments from Referees: Similar comments to Section 4.3 (page 13 lines 5-19). Author's response: Accepted the comment as you suggested. Thank you. Author's changes in manuscript:

25. Comments from Referees: Section 4.3 "157th and 256th". Please give start time (1st January or 1st September) and add the specific time for these two dates, for example 7th (7th January). Author's response: Changed as you suggested. Thank you. Author's changes in manuscript: "Figure 5 shows that the annual mean SODs, SEDs and SDDs are 4th December, 11th March, 99 days, respectively."

26. Comments from Referees: As suggested by Reviewer 1#, the Result section could be shortened. It's helpful to put more attention on snow cover variation results analysis and the new finding interpretation. From Table 3, I find that almost all indexes (SD, SCD,

SOD, SED and SDD) break point occur in the new century (after 2000s). You can give further analysis on what's the different variation rate before and after break point for these indexes. Author's response: Thank you for the comment. Similar responses as Comments 23.

27. Comments from Referees: Section 4.4 in line 21 page 13. change "annual precipitation" to "annual mean precipitation". Author's response: Similar responses as Comments 19. The results from SEM were deleted in the revised paper. Thank you very much. Author's changes in manuscript:

28. Comments from Referees: Page 14 lines 7-12: (Q1): why did the latitude and altitude have different effects on SD_overall and SD_max? "latitude and altitude do no impact SDoverall" but "all factors affect the spatial and temporal distribution of SD-max". (Q2): according to previous studies, latitude and altitude have a significant effect on SD. Your conclusion is "latitude and altitude do not impact SDoverall". Please give me more explanations. Author's response: The latitude, longitude and altitude directly affect precipitation and temperature and thus indirectly affect snow cover, while precipitation and temperature have a direct impact on snow cover. However, the geographical factors only effect on sow spatial distribution, not snow temporal variation. The SEM analysis only found that the air temperature is the major control factor for snow depth and snow phenological changes, and precipitation are only effect for snow depth, and its impact is far less than the temperature. We considered that the SEM analysis results was added little to this paper, thus we decided to delete the part of the SEM. Thank you very much for the comment.

Author's changes in manuscript:

29. Comments from Referees: As we all known, MODIS do not provide SD information. In page 15 line 19-21, "….whereas SD decreased in the north and northwest regions of the Tibetan Plateau from 200 to 2014 according to MODIS snow products". Please revised it. Especially, Figure 3 do not have significant change station in northwest

of the Tibetan Plateau! Author's response: Sorry for our carelessness in discussion. The wrong sentence was revised. Thank you for the comments. Author's changes in manuscript: "Huang et al. (2016) also found a significant increasing trend of SD in Northeast China, whereas SD decreased in the north and northwest regions of the Tibetan Plateau from 2000 to 2014. However, there are no stations shows significant change in northwest of the Tibetan Plateau in the long-term period from 1951 to 2018."

New reference cited in the revised manuscript: Bryant, E.: Climate process & change, UK: Cambridge Univ. Press, 1997. China Meteorological Administration. Specifications for surface meteorological observations. China Meteorological Press, Beijing, 61-63, 2003. Dyer, J. L. and Mote, T.: Spatial variability and trends in observed snow depth over North America, Geophys. Res. Lett., 33, L16503, 2006. Kim, K. Y., and North, G. R.: EOF-based linear prediction algorithm theory, J. Climate, 11, 2046-3056, 1998. Li, D., Wang. C.: Research progress of snow cover and its influence on China climate, Trans. Atmos. Sci., 34, 627-636, 2011. Liu, Y., Peng, G., Chen, X., Yang, Y.: Climatic and environmental changes in Shangri-La in next 50 years according to wavelet analysis and multiple VAR regression prediction modeling, Res. Sci., 38, 1754-1767, http://doi.org/10.18402/resci.2016.09.13, 2016. Ma., N., Yu, K., Zhang, Y., Zhai, J., Zhang, Y., Zhang H.: Ground observed climatology and trend in snow cover phenology across China with consideration of snow-free breaks, Clim. Dyn., 55, 2867-2887, https://doi.org/10.1007/s00382-020-05422-z, 2020. Notarnicola C., Hotspots of snow cover changes in global mountain regions over 2000–2018, Remote Sens. Environ., 243, 111781, 2020. Peng, S., Piao, S., Ciais, P., Fang, J.: Change in winter snow depth and its impacts on vegetation in China, Global Change Biol., 16, 3004-3013, https://doi.org/ 10.1111/j.1365-2486.2010.02210.x, 2010. Percival, D. B., Walden, A. T.: Wavelet Methods for Time Series Analysis (Cambridge Series in Statistical and Probabilistic Mathematics), UK: Cambridge Univ. Press, 2000. Soon, W., Connolly, R., Connolly, M., O'Neill, P., Zheng, J., Ge, Q., Hao, Z., Yan, H.: Comparing the current and early 20th century warm periods in China, Earth-Sci. Rev., https://doi.org/10.1016/j.earscirev.2018.05.013, 2018.

Sun, Y, Zhang, T, Liu, Y, Zhao, W., Huang, X.: Assessing snow phenology over the large part of Eurasia using satellite observations from 2000 to 2016, Remote Sens., 12, 2060, http://doi.org/10.3390/rs12122060, 2020. Xiao, X.; Zhang, T.; Zhong, X.; Li, X. Spatiotemporal Variation of Snow Depth in the Northern Hemisphere from 1992 to 2016. Remote Sens. 2020, 12, 2728. Zhang, X., Wang, K., Boehrer, B.: Variability in observed snow depth over China from 1960 to 2014, Int. J. Climatol., 1-9, https://doi.org/10.1002/joc.6625, 2020. Zhou, C. and Wang, K.: Quantifying the sensitivity of precipitation to the long-term warming trend and interannual–decadal variation of surface air temperature over China, J. Clim., 30(10), 3687–3703. https://doi.org/10.1175/JCLI-D-16- 0515.1, 2017.

---

## Author Comment (AC3) · 1 Nov 2020

Anonymous Referee 1 In this paper the authors' document trends in China snow cover from surface observations of daily snow depth over the period 1951-2018. The paper results are consistent with previous publications showing mainly increasing snow depth and snow cover above 40N, with decreasing snow cover south of 40N. The main merit of the paper is the period of record (1951-2018) which currently represents the most up-to-date (and longest) assessment of snow cover trends in China. The introduction is well-written and comprehensive, but would be improved with more focus and synthesis of the Chinese snow cover literature, and a clearer discussion and presentation of the

study rationale. The data and methods sections are mostly well written, although the methods section could use some additional explanation in a few places (see detailed comments). The trend results are presented clearly, but there is considerable potential to streamline the presentation. The Structural equation modeling component of the analysis is not compelling; it currently lacks a clear rationale and is based on inappropriate air temperature and precipitation variables. The updated trend results presented in the paper are of strong interest to the cryospheric community. However, the paper provides little explanation of the mechanisms responsible for the trends, which is a major weakness.

Author's response: Firstly, on behalf of all authors, we appreciate your careful review and also great comments for this manuscript. Please accept our respect and gratitude to you for your pertinent suggestion and responsible review. Base on your comments, the revised manuscript has made the following changes: 1) The Introduction section was revised based on your comments. Currently various available data for monitoring snow cover observations are referred to, including their advantages and limitation. More literatures focus on snow cover variation in China are cited and discussed. And what issues of snow cover change in China still need to resolve was put forward. 2) We have added more principles description of methods used in the article. Only the climate data in the cold season was re-analyzed in the revised manuscript. We definitely found more interesting results this time. 3) The results of the breakpoint analysis were discussed separately. 4) The structure of the article was re-organized, the results were partially condensed and more discussion has been added in order to explain the mechanisms responsible for the trends of snow cover in China. Include an enhance correlation analysis between climate and snow cover, snow cover spatiotemporal pattern based on EOFs analysis, as well as the oscillation cycle based on Morlet wavelet.

Detailed comments: 1. Comments from Referees: Suggested wording change for first line of Abstract: "Snow cover changes over China from 1951 to 2018 are documented based on an analysis of in situ daily snow depth observations from 730 meteorological

stations. The snow cover indicators analyzed included snow depth (SD), snow covered days (SCDs), and snow phenology."

Author's response: Changed as you suggested. Thank you.

Author's changes in manuscript: In Abstract: "Snow cover changes over China from 1951 to 2018 are documented based on an analysis of in situ daily snow depth observations from 730 meteorological stations. The snow cover indicators analyzed included snow depth (SD), snow covered days (SCDs), and snow phenology."

2. Comments from Referees: The Introduction is well written and comprehensive, but it needs to focus more on China snow cover. I think you could delete the first two paragraphs and replace this with a focussed discussion of the various advantages and disadvantages of the currently available data for monitoring snow cover changes over China. In this regard, your statement that in situ snow depth observations provide the most reliable dataset for analyzing snow cover changes "with a high degree of credibility" will need to acknowledge the strong local scale processes influencing point snow depths, the poor spatial distribution of stations in some regions of China, and the low-elevation bias in the station network. A summary table of China snow cover trend results from previous studies would be a useful addition to the Introduction given the sensitivity of trend results to the specific period of data analysed. The recent findings by Ma et al. (2020) of the role of changes in winter snow-free periods in snow cover duration trends should be included in the discussion as they help explain why snow cover duration can increase under warming temperatures. A concise synthesis of previous results and identification of knowledge gaps is important for providing a clearer rationale for this particularly study. For example, the SEM analysis presented in subsection 3.2 appears to be an innovative aspect of the paper that needs to be incorporated in the study rationale.

Author's response: We completely agree with the comment. In the revised manuscript, more focus on previous studies on China snow cover was added in the introduction section. The limitation of the poor spatial distribution of stations in the west of China was discussed. Due to many of literature focus on the snow cover change in regional scale in China, especially the data and methods are different, only a few of literatures refer to throughout the China snow at present. It is difficult to compare different conclusions in the form of tables. Therefore, we have set aside a section in discussion section to compare with previous studies, include the latest literature. The SEM analysis are indeed added little to the paper. After discussion, the authors decided to delete the part of the SEM analysis results. Thank you very much for your great advice. Author's changes in manuscript: In Introduction section: "Limited by the coarse spatial resolution from passive microwave remote sensing data and severely cloud obscured from optical remote sensing data, in situ snow observations provide the most reliable dataset for analyzing the changes in snow cover with a high degree of credibility. Moreover, snow parameters are calculated from meteorological station data, which have great advantages in the process of long time series research. However, in situ observations from climate station is insufficient for representing at a regional scale due to spatial discontinuities, irregularities and inhomogeneities in the distribution. Most stations distributed in the flat and open area, and rare distribution in west China, especially the Tibetan Plateau. The stations are lacking in inner and mountain areas in plateau, greatly limits the observation integrity and representation of space. Nevertheless, the sparse station network still provides the long-term of high-quality ground observation data."

3. Comments from Referees: What is your definition of "stable" snow cover? Are TP, NX and NC highlighted in Figure 1 because they are the only areas in China with a stable snow cover? Is this also the reason that only these three regions are summarized in the results? The issue of ephemeral vs stable snow cover deserves some discussion particularly in light of the Ma et al. (2020) paper. Related to this, your statistical analyses are carried out at all stations in China not just those with a stable snow cover. How robust are the statistical methods in ephemeral snow cover areas with frequent zero snow cover years and undefined start and end dates to the snow season?

Author's response: The stable snow cover area means the area with the mean annual snow-covered days bigger than 60d during the snow season. The stable snow cover only distributes in TP, NX, and NC (Huang et al. 2016; 2017). In this study, the stations are distributed throughout the China are used to access the snow cover changes during the period of 1951 to 2018. The TP, NX and NC highlighted and discussed in this study, the main reason is that the snow in these three regions has more significance in regional climate, hydrology and ecology compare to other regions. We agree the ephemeral snow also play important role in climate change, thus in the revised manuscript, the summary and description are as inclined as possible to the whole of China, not only the TP, NX and NC, but also the ephemeral snow areas. To ensure consistent statistical results between the stable and ephemeral snow areas, strict statistics of snow cover indices was adopted. For example, to avoid the impact of ephemeral snow in snow phenology computations, SOD was defined as the first date of the first three continuous snow records, and SED was defined as the last day of the date of the last three continuous snow records in a hydrological year (from 1st July to 30th June of the next year). If the rules are not met, such as in ephemeral snow cover areas with zero or less than three continuous snow records in snow cover years, the statistics of the year will be abandoned.

4. Comments from Referees: In Section 2.2 the use of an annual period seems strange given the snow season is confined to a much shorter cold season. It is also not clear how annual maxima of air temperature and precipitation would assist in diagnosing changes in snow cover. From energy and mass balance considerations variables like freezing degree-days, total solid precipitation and the solid-fraction of total annual precipitation would be expected to have more relevance for explaining variability and change in snow cover.

Author's response: We completely agree with the comment. It is thoughtless to use annual period meteorological data to explain the cause of snow cover variation. In the revised paper, we only choose the temperature and precipitation in the cold season

to analyze the response of snow cover to climate change. We think the maximum temperature in the clod season may accelerate the snow melting, especially in snow melt season, which may cause the snow depth to become shallower and the end date get earlier. To correspond to the snow season, the temperature and precipitation in clod season are selected this time. Thank you for your advice. Author's changes in manuscript: In 2.2 Meteorological data section "The dataset time series is from 1901 to 2017. In this study, the monthly mean temperature (Tmp-mean), minimum temperature (Tmn-min), maximum temperature (Tmx-max), and precipitation in the cold season (October to March of the following year) from 1951 to 2017 in the dataset were used to explore the spatiotemporal heterogeneity of snow cover variation."

5. Comments from Referees: Section 3.1: Can you provide a line or two of text prior to eqns. 3 to 6 to explain what these equations are being derived for? I suggest you add a new section "3.2 Change-point analysis". Overall I find section 3.1 a bit confusing and statements in the Results section further increase my confusion e.g. page 9 line 4 "The results of the M-K trend test are the same as those of the slope method".

Author's Response: Thank you for the comments. More explanation was added to those equations, and a new section for the breakpoint test was added. The M-K trend test was deleted, the method only employed for the breakpoint test. And the slope method was employed to analyze the trend of snow cover at each station as well as the overall snow cover trend from 1951 to 2018 in China, respectively.

Author's changes in manuscript: In "3.2 Breakpoint test" section "The Mann-Kendall (M-K) test is recommended by the World Meteorological Organization and is frequently used to analyze the trends of changes in meteorological and hydrological elements (Milan, 2013). M-K test can not only check the variation overall trend of the sequence but also specify the breakpoint starts. First, for a time series x with n samples, a sequential column Sk was constructed. Under the assumption of random independence of time series, a sequence of statistic UF was defined. Then, in reverse order according to time series x, the above process was repeated to calculate UB. The method is realized

by the following formula: where Sk is the cumulative count of the number of values at the time i great than at time j. E(Sk) and var(Sk) are mean and variance values of Sk. UF is the standardized value of S, while UB is obtained by inverting the sequence of UF. "

6. Comments from Referees: Section 3.2: Please provide some introductory text to your current section 3.2 on why you proposing to employ SEM? What are the hypotheses you are testing and why is SEM the best method? In your statement that "seven factors were screened out", I think you mean that seven factors were retained for analysis. As mentioned previously, the use of annual maxima in this analysis is difficult to justify for understanding snow cover variability. I think you would learn more about snow cover variability by defining the regional snow cover response regions from EOF analysis, then looking at the corresponding regional time series of snow season air temperature, total precipitation, and precipitation solid fraction.

Author's Response: Thank you for the comments and suggestions. The comments for SEM was responded in Author's response 2. Based on your suggestion, the EOFs was employed in the revised manuscript to reveal the spatiotemporal pattern of snow cover, further proved that temperature is the main driving factor of snow cover variation in time and space. Author's changes in manuscript: Please see section 5.3 Spatiotemporal pattern and driving factors of snow cover variation.

7. Comments from Referees: Include slope units in Table 2. Why is the China average not included as in Table 3? The same applies to the trend result tables for other snow cover variables. Author's response: Revised as you suggested. Thank you very much.

8. Comments from Referees: Can you explain how the anomaly time series in Figure 4 is obtained? There should be an error bar for each annual mean, and the error will influence the linear fit through the points. Can you also provide a brief explanation how to interpret the UF and UB curves in Figures 4b and 4d. Wouldn't the confidence interval in the trend get narrower as the length of the time series increases?

Author's response: In M-K analysis, only the interannual variation of the mean value can be used to test the breakpoint. Figure 4a was intended to show the overall trend of average snow depth. The UF and UB interannual variation curves are the key results for revealing the snow cover variation. The borderlines in the figure mean the UF and UB equal to $\pm1.96$, the trend reaches to significant level (P <0.05). The confidence interval is constant, don't get narrower as the length of the time series increases. In the revised manuscript, the means of UF and UB were explained in the methodology section, and the UF curves for each snow indices were also introduced. Thank you for the comments.

Author's changes in manuscript: In section "4.2.2 Breakpoint test" Figure 6 shows the trend of different snow cover indexes changing with time during the period from 1951 to 2018 in China. From the perspective of time variation trend, each snow cover index has its own characteristics over time, which does not change linearly, but shows an increasing or decreasing trend of fluctuation and oscillation. The overall trend of snow depth is increasing, but the change has roughly experienced the process of first increasing, then decreasing, and then increasing again. The trend of SDoverall changes in 1961 and 1997, and increase significantly after 2010. The trend of SDmax changes in 1958 and 2007, however, the overall increase was not significant. SDDs represents the length of the snow season, while SCDs represents the number of days the surface is covered by snow. During the period of 1951 to 2018, the SCDs has experienced three processes of increasing first, then decreasing, and then increasing, with the overall increasing trend but not significant. While the SDDs first increase before 1999 and then decrease, and reduction significantly after 2015, indicating that the shortening of snow season in recent years is a well-established fact. The change of SOD is relatively complex. The SOD experienced several trends of postponing or advancing, and the overall trend was delayed but not significant. SED showed a trend of delay until 1998, but then getting earlier, and advance significantly after 2014. Table 3 summarized the breakpoints of six snow cover indices detected by a moving t test. The results indicate that the spatiotemporal variations in snow cover show obvious regional differences.

Specially, there is no significant breakpoint in the Tibetan Plateau. Furthermore, the snow phenology changes in the Tibetan Plateau are greater than those elsewhere. The abrupt change years of snow cover in different regions were almost different. However, almost all the abrupt change occurred around the 2000s, indicating that snow cover had changed significantly in the 20th century in China since 1951.

9. Comments from Referees: What is responsible for the break points shown in Table 3? Are they linked to changes in atmospheric circulation?

Author's response: If the UF and UB curves intersect between the borderlines (u0.05 = ±1.96), the point of intersection corresponds to the time at which the breakpoint transition begins. However, the breakpoints of the meteorological sequence can be further judged by combining the M-K and moving t tests. When t is greater than t0.05, the year corresponding to t represents a breakpoint. Our study found that the long-term trend of snow cover variation is mainly due to warming climate in the cold season, and the short-term anomaly of snow cover is related to extreme weather, such as the abnormal increase of snow depth in negative temperature anomaly.

10. Comments from Referees: Section 4.4: The results of the SEM analysis are not very convincing and add little to the paper. The analysis may be more instructive using air temperature and precipitation variables that are more closely linked to snow cover variability.

Author's response: We completely agree with the comment. The SEM analysis is indeed added little to the paper. After discussion, the authors decided to delete the part of the SEM analysis results. Instead with an enhanced correlation analysis between climate and snow cover variation in the Discussion section. Thank you for the comments.

11. Comments from Referees: I think your Results section could be significantly shortened if you presented all the snow cover variables together instead of separately. I think this would also help interpreting and explaining the results. At the moment the

results are presented in a rather descriptive way following the same format for each variable, which is not very interesting from the readers point of view.

Author's response: In the revised version, the structure of the article was re-organized based on the comments. Thank you so much. Author's changes in manuscript: The catalog in the revised manuscript is the following: 1 Introduction 2 Dataset 2.1 Snow depth records 2.2 Meteorological data 3 Methodology 3.1 Trend analysis 3.2 Breakpoint test 4 Results 4.1 Spatiotemporal characteristics of snow cover in situ observation 4.1.1 Snow depth 4.1.2 SCDs 4.1.3 Snow phenology 4.2 Snow cover trends across China 4.3 Breakpoint 5 Discussion 5.1 Compared with previous study 5.2 Snow cover related to climate change 5.3 Spatiotemporal pattern and diving factors of snow cover variation 5.4 Dose the oscillation cycle exists in snow cover varation? 6 Conclusion

12. Comments from Referees: The conclusions are largely descriptive and it is hard to find any new insights into snow cover variability and change in China in this paper. As it stands, the only significant contribution of the paper is to extend the period of previous trend analyses. I see several areas where the authors could make potential new contributions: - document the snow response regions of China from EOF analysis of station annual series of SDmax and SCD series - determine the roles of regionally-averaged (over the identified snow response regions) snow season air temperature, total precipitation, and total snowfall in the observed snow cover series. - find physical explanations for break points e.g. atmospheric circulation changes, increased snowfall in winter storms, fewer snow-free periods (e.g. Ma et al. 2020). - find physical mechanisms for the decrease in snow season gaps documented by Ma et al. (2020)

Author's response: The conclusion section was rewrote based on your suggestions. The main conclusions of this study are summarized, and the prospect of future research is put forward according to the new findings in this study. Thank you.

Author's changes in manuscript: In section "6 Conclusion" "While ground measurements are the most accurate way for plot-scale snow information, the results reported

in the current study would bear high uncertainties in poorly monitored areas, especially in the Tibetan Plateau, in which stations are much less when compared to eastern China. Here, long-term snow cover variations were assessed using in situ observations across mainland China from 1951 to 2018. The variation of snow cover has obvious regional deference. Snow depth tends to increase northward of 40°N, but decrease southward of 40°N, the overall snow depth increases significantly in China, where the greatest contribution came from the increase in north China. SCDs in Northeast China and northern Xinjiang both show increasing trends, especially in Northeast China, where the increase is significant. The snow season tends to shorten throughout mainland China, jointly resulting from later snow onset and earlier snow melting, which mainly due to the warming climate in clod season. Furthermore, the snow phenology changes in the Tibetan Plateau are greater than those elsewhere. China began warming persistently after 1980s and warmed significantly after the 1990s since 1951. With climate warming, snow depth, and its phenology change severely, and temperate is highly related to the change of snow cover. However, the temperature significantly negatively correlated with snow depth, the snow depth increased significantly in north China mainly possible due to the increase in temperature strengthen the south-north airflow exchange and water vapor circulation, thus increase of snowfall events in northern China and alpine areas. After the 2000s, the warming gradually slowed down, the unusually low-temperature years finally led to a significant trend of increasing snow depth. In addition, the abrupt change of variation occurred around the 2000s, indicating that snow cover had changed significantly in the 20th century in China since 1951. Furthermore, the snow cover varies interannual, the snow depth fluctuates, and changes periodically with the increase of temperature. The long-term trend of increasing snow depth is caused by warming in the cold season, and the short-term anomaly of snow depth is related to extreme weather, such as the abnormal increase of snow depth in negative temperature anomaly. The change of snow phenology is mainly caused by the warming climate in the cold season. This study also shows that the influence of the temperature effect on snow cover is more important than precipitation obviously.

With climate change, especially climate warming, snow depth, and its phenology will change severely, which will dominate by the temperature rise in the future. The large-scale fluctuation of snow cover must be the result of climate change, and the analysis of abnormal events in the weather system is a way to further reveal the interannual variation and regional differences of snow cover. However, this study also found that the abrupt change of snow cover occurred mainly around 2000, but the temperature warming gradually slowed down after 2000 in China. What causes the abrupt change of snow cover needs to be further explained in combination with extreme weather, atmospheric circulation, etc."

New reference cited in the revised manuscript: Bryant, E.: Climate process change, UK: Cambridge Univ. Press, 1997. China Meteorological Administration. Specifications for surface meteorological observations. China Meteorological Press, Beijing, 61-63, 2003. Dyer, J. L. and Mote, T.: Spatial variability and trends in observed snow depth over North America, Geophys. Res. Lett., 33, L16503, 2006. Kim, K. Y., and North, G. R.: EOF-based linear prediction algorithm theory, J. Climate, 11, 2046-3056, 1998. Li, D., Wang. C.: Research progress of snow cover and its influence on China climate, Trans. Atmos. Sci., 34, 627-636, 2011. Liu, Y., Peng, G., Chen, X., Yang, Y.: Climatic and environmental changes in Shangri-La in next 50 years according to wavelet analysis and multiple VAR regression prediction modeling, Res. Sci., 38, 1754-1767, http://doi.org/10.18402/resci.2016.09.13, 2016. Ma., N., Yu, K., Zhang, Y., Zhai, J., Zhang, Y., Zhang H.: Ground observed climatology and trend in snow cover phenology across China with consideration of snow-free breaks, Clim. Dyn., 55, 2867-2887, https://doi.org/10.1007/s00382-020-05422-z, 2020. Notarnicola C., Hotspots of snow cover changes in global mountain regions over 2000–2018, Remote Sens. Environ., 243, 111781, 2020. Peng, S., Piao, S., Ciais, P., Fang, J.: Change in winter snow depth and its impacts on vegetation in China, Global Change Biol., 16, 3004-3013, https://doi.org/ 10.1111/j.1365-2486.2010.02210.x, 2010. Percival, D. B., Walden, A. T.: Wavelet Methods for Time Series Analysis (Cambridge Series in Statistical and Probabilistic Mathematics), UK:

Cambridge Univ. Press, 2000. Soon, W., Connolly, R., Connolly, M., O'Neill, P., Zheng, J., Ge, Q., Hao, Z., Yan, H.: Comparing the current and early 20th century warm periods in China, Earth-Sci. Rev., https://doi.org/10.1016/j.earscirev.2018.05.013, 2018. Sun, Y, Zhang, T, Liu, Y, Zhao, W., Huang, X.: Assessing snow phenology over the large part of Eurasia using satellite observations from 2000 to 2016, Remote Sens., 12, 2060, http://doi.org/10.3390/rs12122060, 2020. Zhang, X., Wang, K., Boehrer, B.: Variability in observed snow depth over China from 1960 to 2014, Int. J. Climatol., 1-9, https://doi.org/10.1002/joc.6625, 2020. Zhou, C. and Wang, K.: Quantifying the sensitivity of precipitation to the long-term warming trend and interannual–decadal variation of surface air temperature over China, J. Clim., 30(10), 3687–3703. https://doi.org/10.1175/JCLI-D-16- 0515.1, 2017.